# Host ecology and phylogeny shape the temporal dynamics of social bee viromes

Vincent Doublet [1,2] ✉, Toby D. Doyle [2], Claire Carvell [3],
Mark J. F. Brown [4] & Lena Wilfert [1,2]

The composition of viral communities (i.e. viromes) can be dynamic and complex. Co-evolution may lead to virome host-specificity. However, eco-evolutionary factors may influence virome dynamics in wild host communities, potentially leading to disease emergence. Social bees are relevant models to address the drivers of virome composition: these important pollinators form multi-species assemblages, with high niche overlap and strong seasonality in their biotic interactions. We applied a microbial community approach to disentangle the role of host phylogeny and host ecology in shaping bee viromes, combining plant-pollinator networks with meta-transcriptomics, and small interfering RNAs as proxies for viral replication in pollinators and pollen. We identified over a hundred insect and plant viral sequences from ca. 4500 insect pollinator samples across three time points in one year. While host genetic distance drives the distribution of bee viruses, we find that plant-pollinator interactions and phenology drive plant virus communities collected by bees. This reveals the opportunities for virus spread in the bee assemblage. However, we show that transmission to multiple hosts is only realized for a fraction of insect viruses, with even fewer found to be actively replicating in multiple species, including the particularly virulent multi-host acute bee paralysis virus.

Viruses are obligate intracellular, often pathogenic microbes. They have important ecosystem functions, regulating host populations and selecting the most resistant lineages[1]. The assemblage of viruses transiently carried by or infecting one host and its microbiota is defined as an organism's virome[2]. Host viromes represent complex and highly dynamic viral communities composed of phylogenetically diverse species with different epidemiology, tissue tropism, and transmission routes[3]. Because of the high dependency of viruses on the host cell replication machinery, viruses are often host specific[4,5]. Additionally, ecological factors may also influence the composition of viromes, such as host contact opportunities, and their seasonal variations, leading to pronounced spatio-temporal dynamics[6,7] and a high potential for host switching[8]. Despite the ecological impact of viromes on their hosts and ecosystems, the combined effect of these eco-evolutionary factors on viral assemblage has rarely been explored in

insect communities. Studies primarily focusing on vertebrate hosts found a predominant role of host phylogeny on virome composition, and a limited, but sometimes significant effect of habitat overlap[7,9–11] or inter-species interactions such as predator-prey relationships[4] increasing cross-species virus transmission likelihood, particularly among phylogenetically related species[12,13]. Here, we followed the temporal dynamics of insect pollinator viromes, exploring the relative roles of host phylogeny and plant-insect interactions on the dynamics of viruses.

Wild and managed bees represent particularly relevant models to study virome dynamics[14,15]. They form multi-species and highly connected assemblages within the community of insect pollinators, with a high potential for horizontal cross-species viral transmission via the shared use of floral resources[16]. They include phylogenetically close social species (i.e., Apinae), which show strong seasonal size variation,

[1]Institute of Evolutionary Ecology and Conservation Genomics, University of Ulm, Ulm, Germany. [2]Centre for Ecology and Conservation, University of Exeter, Penryn, UK. [3]UK Centre for Ecology & Hydrology, Wallingford, UK. [4]Department of Biological Sciences, School of Life Sciences and the Environment, Royal Holloway University of London, Egham, UK. ✉e-mail: vincent.bs.doublet@gmail.com

with colonies consisting of up to thousands of non-reproductive individuals in summer, but undergoing strong winter bottlenecks. Bee viromes are primarily composed of single-strand RNA viruses, that are epidemiologically and phylogenetically diverse[17–19], including several multi-host viral species[20,21]. Many eco-evolutionary factors influence disease dynamics in bees, with patterns of virus or microparasite prevalence affected by host traits[22] and taxonomy[23–25], transmission route[26,27], seasonality and geography[17,28], host density[29,30] and management[31], host community composition[32–34] and transmission opportunities driven by floral diversity and abundance[33–36]. However, these effects have been investigated mainly in single host-pathogen systems, whilst bee viruses evolve in multi-host landscapes[33,37–39]. Viral diseases also represent a serious threat to honeybees and beekeeping. In particular, deformed wing virus (DWV)[40–42] and sacbrood virus (SBV)[43] may induce unsustainably high individual and colony mortality in the western (*Apis mellifera*) and Asian (*Apis cerana*) honeybees, respectively.

Only a handful of studies have characterized the virome of sympatric bee species, and the drivers that shape their composition. Pascall et al.[24] showed that phylogenetically related bumblebee species are infected at similar frequencies by the same sets of viruses, whilst Robinson et al.[44] found little overlap in virome composition across a wider phylogenetic diversity of sympatric social and solitary bees. A role for ecological factors such as niche overlap between sympatric bee species, as well as its temporal dynamics, has been hypothesized[14,45], but never investigated at the whole virome scale.

Here, we applied a microbial community approach to understand the role of ecological, evolutionary, and temporal factors on the virome dynamics of common pollinators. We collected the most prevalent insect pollinator species, importantly incorporating both wild and managed bees and two other pollinator groups that are dominant in the northern hemisphere, hoverflies[46] (Syrphidae spp.) and the so-called "forgotten flies"[47] (i.e., non syrphid dipteran), from ten farms in Southern England across three time points in 1 year. We sequenced their meta-transcriptomes and small RNAs to identify and discover RNA viruses. In combination with a temporal analysis of plant-pollinator networks, we tested the effect of host phylogeny and foraging niche on sympatric hosts' viromes. We hypothesize that infectious bee viruses will be largely restricted to their primary host, while the presence of non-infectious viruses, such as plant viruses picked up while foraging, will be driven by the seasonal variation of ecological interactions within plant-pollinator networks.

## Results

### Bee viruses are mainly host-specific, plant viruses cluster by season

We sequenced 16 libraries to characterize the RNA virus communities from social bees and other dominant insect pollinators caught in ten farms in Southern England, across three time points (Fig. 1). Thirteen of these libraries were generated from a single species and time pool of RNA (i.e., *A. mellifera*, *Bombus hortorum*, *Bombus lapidarius*, *Bombus pascuorum* and *Bombus terrestris*) and the other 3 from pools of bee or fly species (Supplementary Data 1). Transcriptome analysis resulted in the assembly and identification of 143 viral Operational Taxonomic Units (OTUs), including 39 plant viruses, identified via sequence homology. Picornavirales represented by far the most abundant viruses in all bee viromes (Supplementary Fig. 1).

After normalizing read counts mapped on viral genome assemblies, we measured viral species richness and Shannon diversity from insect and plant viruses separately. We found honeybees to carry more insect viruses in summer than bumblebees (June $\chi^2 = 18.54$, df = 4, $p = 0.002$; August $\chi^2 = 16.87$, df = 4, $p = 0.002$, but not in April $\chi^2 = 2.27$, df = 2, $p = 0.321$) although alpha diversity was not different between the two groups (Kruskal-Wallis test $\chi^2 = 0.46$, df = 1, $p = 0.499$; Fig. 1a). We examined the differences of pollinator virome compositions with a

multivariate analysis on Bray-Curtis distances using host taxonomic groups and collection time points as key factors. Virome composition with insect viruses was significantly structured by host taxonomic group (PERMANOVA $F = 23.485$, $R^2 = 0.93$, df = 7, $p < 0.001$; Fig. 1b), and to a lesser extent by collection time point ($F = 3.043$, $R^2 = 0.03$, df = 2, $p = 0.002$). This result is not affected by the viromes from mixed time points (i.e., *Andrena* spp., hoverflies, and the forgotten flies), as the same result is obtained when they are discarded, with both host taxonomy ($F = 23.517$, $R^2 = 0.89$, df = 4, $p < 0.001$) and collection time points remaining significant ($F = 3.043$, $R^2 = 0.06$, df = 2, $p = 0.015$). The composition in plant viruses was structured both by collection time point ($F = 13.327$, $R^2 = 0.50$, df = 3, $p < 0.001$), with a tight cluster of samples collected in April in a two-dimensional graphical representation (Fig. 1c), and by insect taxonomic group ($F = 5.533$, $R^2 = 0.42$, df = 6, $p < 0.001$). This pattern is also observed when viromes from mixed time points are removed from the analysis, with both collection time points ($F = 12.886$, $R^2 = 0.57$, df = 2, $p < 0.001$) and host taxonomy remaining significant ($F = 3.449$, $R^2 = 0.30$, df = 4, $p = 0.002$). This is further illustrated by performing two-way cluster analyses. Viromes restricted to insect viruses clustered mainly by host taxa (Fig. 1d). Honeybee viromes, across time points, segregated out from all other samples; they were characterized by a high abundance of common viruses such as black queen cell virus (BQCV), DWV genotypes A and B, and SBV, as well as a large diversity of Lake Sinai viruses and *Apis* rhabdoviruses only detected in honeybees. Bumblebee viromes clustered together, sharing high levels of slow bee paralysis virus (SBPV), acute bee paralysis virus (ABPV), and Mayfield virus 1. Within this group, libraries clustered mainly by species, and not by collection time. Two libraries of fly samples, composed of hoverflies and all other dipteran flower visitors, named here the "forgotten flies"[47], respectively, cluster with a library generated from solitary mining bees (*Andrena* sp.), with a smaller overlap with viromes from bumblebees and honeybees.

In contrast, viromes composed of plant viruses mainly clustered by collection time point (Fig. 1e). These clusters largely reflect the blooming periods of putative host plants, as illustrated by the cluster of samples collected in April showing high levels of Cherry virus A and *Prunus* virus F, likely collected by bees from cherry and plum trees typically blooming in spring in Europe. A second cluster of samples from June mainly shows high levels of cryptic plant viruses, bean yellow mosaic virus, and *Dulcamara* mottle virus. The third cluster, consisting of samples collected in August, shows higher levels of strawberry latent ringspot virus variants, red clover nepovirus A, and other cryptic plant viruses. Other plant viruses such as white clover cryptic virus 2 (WCCV2), infecting clover species in bloom for a long period of time, and the generalists *Arabis* mosaic virus (ArMV) and raspberry ringspot virus showed no temporal pattern in their distribution profile.

### A limited number of bee viruses infect multiple host species

We used small RNA sequencing to determine which viruses showed signs of active replication in our samples, as a proxy for infection. Importantly, viral-derived small interfering RNA (vsiRNA) signals were found not only against insect viruses, but also against plant viruses. While bee RNA interference (RNAi) response typically produces vsiRNA profiles with a majority (i.e., peak) of 22 nt long fragments[48], plant RNAi generally produces a diversity of profiles with a majority of reads around 21–22 nt long fragments[49,50]. Plant RNAi activity against plant viruses detected in our insect samples likely originated from ingested or carried pollen grains. Overall, we recovered 50.9% and 55.8% of insect viruses identified by meta-transcriptomics in honeybees and bumblebees, respectively, with small RNA sequencing (Fig. 2a, Supplementary Data 2). Analyzed at the host species level, we found 24 insect viruses (25%) to be multi-host using this method, i.e., with vsiRNA evidence of replication in more than one host species

**a**

| Host species | A. mellifera | | | B. terrestris | | | B. lapidarius | | | B. hortorum | | B. pascuorum | | Andrena spp. | Hoverflies | Forgotten Flies |
|---|---|---|---|---|---|---|---|---|---|---|---|---|---|---|---|---|
| Collection time | April | June | August | April | June | August | April | June | August | June | August | June | August | Mix | Mix | Mix |
| Insect virus richness | 36 | 47 | 45 | 26 | 35 | 26 | 26 | 25 | 25 | 22 | 24 | 18 | 16 | 28 | 17 | 15 |
| Insect virus Shannon | 1.05 | 0.93 | 1.41 | 1.93 | 0.79 | 0.98 | 1.69 | 1.38 | 0.74 | 1.21 | 1.22 | 1.54 | 1.76 | 2.22 | 1.27 | 0.83 |
| Plant virus richness | 9 | 26 | 16 | 10 | 19 | 7 | 10 | 16 | 13 | 8 | 9 | 13 | 7 | 13 | 15 | 2 |
| Plant virus Shannon | 1.59 | 1.04 | 0.5 | 1.07 | 1.30 | 0.37 | 0.96 | 0.87 | 0.86 | 1.85 | 0.61 | 1.69 | 1.01 | 2.26 | 1.91 | 0.58 |

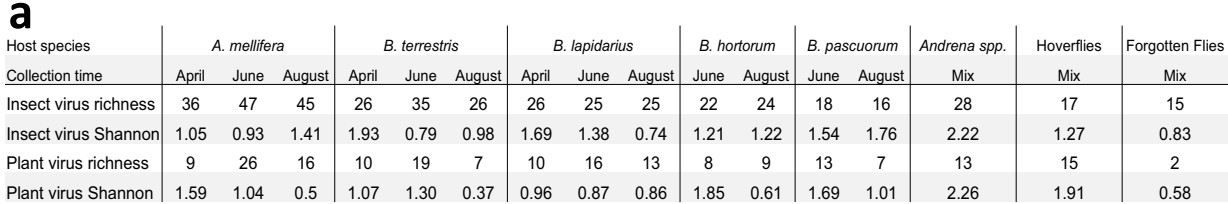

**Fig. 1 | Virome diversity measures and composition. a** Species richness and Shannon diversity index measured for all viromes, restricted to insect viruses, and plant viruses. **b**, **c** Show non-metric multidimensional scaling (NMDS) plots projecting the Bray-Curtis distance matrices of virome comparison across samples using insect and plant viruses, respectively. **d** Heatmaps and cluster analysis of viromes restricted to insect viruses, and **e** plant viruses. Normalized viral read counts are shown on a log scale. Source data are provided as a Source Data file.

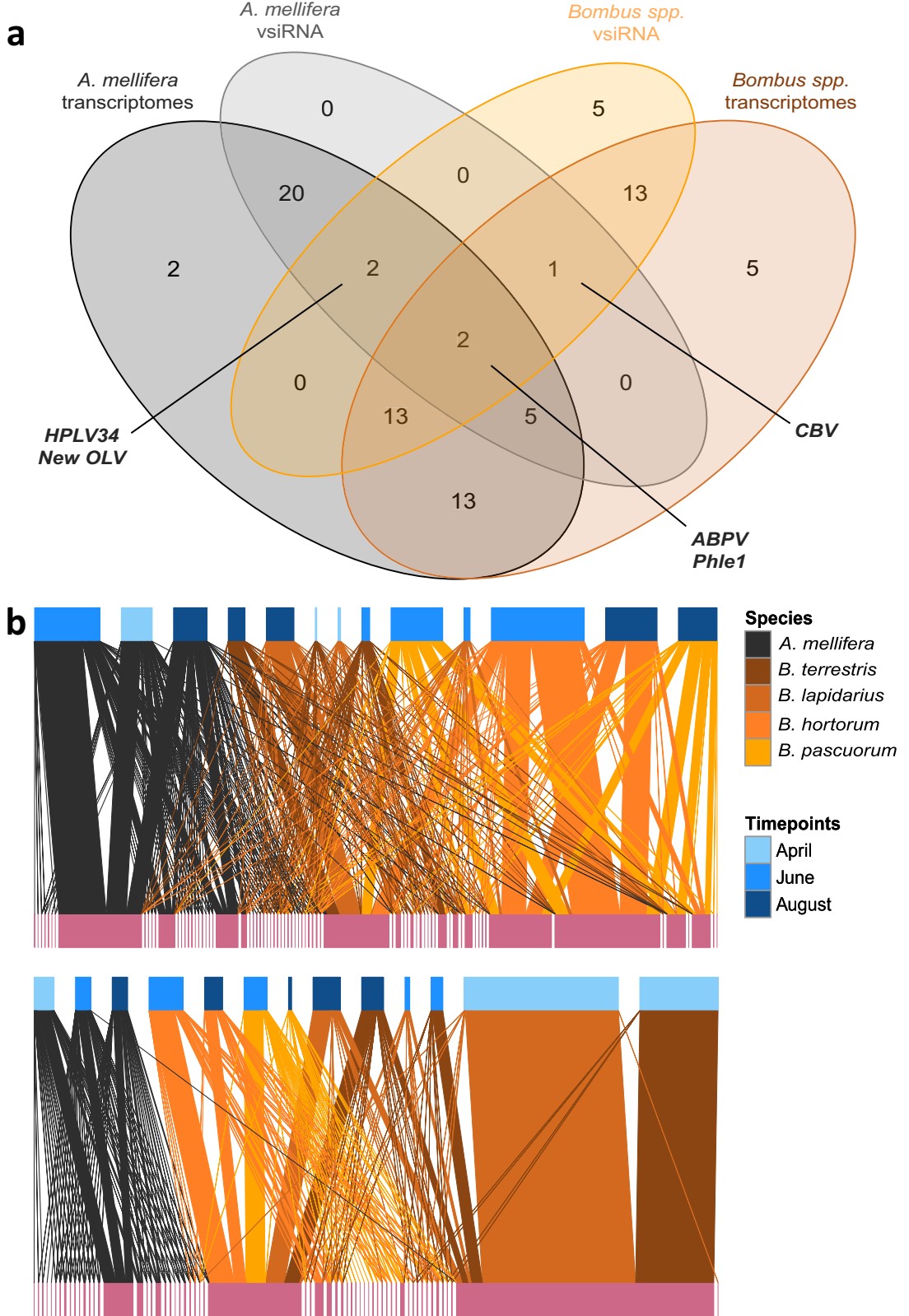

**Fig. 2 | Comparison of meta-transcriptomes and vsiRNAs. a** Venn diagram showing the number of insect viruses detected from meta-transcriptomes and small RNA sequencing in honeybees and bumblebees. Only five insect viruses were shown, by this proxy, to be replicating in both honeybees and bumblebees: Acute bee paralysis virus (ABPV), Bombus associated virus Phle1, Castleton Burn virus (CBV), Hubei partiti-like virus 34 (HPLV34) and a new Osugoroshi-like virus (New OLV). The lists of viruses with vsiRNA signal and their coverage maps are available in Supplementary Data 2. **b** Comparison of networks showing the interactions between social bees (higher nodes) and insect viruses (lower nodes) from meta-transcriptomes (top network) and from vsiRNAs (bottom network), showing a stronger compartmentalization between honeybees and bumblebees in the latter. Higher nodes are colored according to collection time points (shades of blue), edges are colored according to the host species and lower nodes (viruses) are all pink. Source data are provided as a Source Data file.

(Supplementary Figs. 2 and 3). Among them, only five viruses showed vsiRNA signals both in *A. mellifera* and bumblebees: ABPV, which shows evidence of replication in all bumblebees, Castleton Burn virus, mainly replicating in bumblebees but for which a few vsiRNAs are found in honeybees collected in June, the Bunyavirales *Bombus* associated virus Phle1, and two viruses replicating in honeybees and the bumblebee *B. hortorum*, namely Hubei partiti-like virus 34 (HLPV34) and a new Osugoroshi-like virus (New OLV). For DWV and BQCV, which were identified in most meta-transcriptomes, vsiRNAs were only found in honeybees. In contrast, we did not find vsiRNA reads for SBPV in honeybees, while it was present in all bumblebees. Overall, vsiRNA provides a highly sensitive detection method for viruses[51], accordingly we detected seven viruses in the bumblebee vsiRNA data that were not found in the meta-transcriptomes (HLPV34 and New OLV, plus New Bee Iflaviridae 1; New Castleton Burn-like virus; New Jingchuvirales, New Rhabdoviridae 4; New Totiviridae 2) despite our conservative detection threshold. Interestingly, *Varroa destructor* virus 2 (VDV-2) shows 23 nt long vsiRNAs (Supplementary Fig. 3), which is a potential signature of the *V. destructor* RNAi response[52] in our honeybee samples from June. Overall, network projections of insect-virus patterns generated from meta-transcriptomes and vsiRNA in social bees show that honeybees and bumblebees appear to share only part of their virome, including few actively replicating viruses, while bumblebees share a larger number of replicating viruses, particularly in the summer months (Fig. 2b). Both networks showed high levels of modularity $Q$ in comparison to random (null) networks (meta-transcriptome network: $Q = 0.52$; $\bar{Q}_{null} = 0.0017$, sd = 0.0002, z-score = 2517; vsiRNA networks $Q = 0.51$; $\bar{Q}_{null} = 0.02$, sd = 0.0017, z-score = 279), illustrating the high level of specialization of virome composition across bee species. Interestingly, we detected a very strong presence of Castleton Burn virus in vsiRNAs from *B. lapidarius* and *B. terrestris* spring queens.

## Host phylogeny, host niche, and plant phenology define virome composition

Using meta-transcriptomes from single-species pools only (i.e., discarding species pools for *Andrena* spp., hoverflies, and the "Forgotten flies"), we examined the effects of host phylogenetic distance, host ecological distance, and collection time points on the beta-diversity of social bee viromes, and found contrasting effects for insect and bee-associated plant viruses. For insect viruses, only host phylogenetic distance explained virome composition (Likelihood Ration Test (LRT): $\chi^2 = 9.222$, $p = 0.002$), with higher genetic distance associated with more dissimilar viromes (Fig. 3a, c). The phylogenetically distant honeybees show more dissimilar viromes as compared to the closely related bumblebees, both in June and August, even though virome distance is discordant with phylogenetic relationships in the bumblebee clade in June (Fig. 3e). In contrast, the virome composition of the identified plant viruses was explained by the interaction between bees' ecological niche distance, measured from plant-pollinator networks, and sample collection time points (LRT: $\chi^2 = 11.454$, $p = 0.003$) (Fig. 3b, d). We also found a seasonal effect of virome dissimilarity: plant viruses were shared more across bee species in August than in June (LMM: $t = -2.29$; $p = 0.022$).

## Discussion

In this study, we combined comparative meta-transcriptomics, small RNA sequencing, and ecological network analysis to identify drivers of virome composition in insect pollinators, following on from our study of the transmission dynamics of three key bee viruses (DWV-A, DWV-B, and ABPV) at the individual level in this population[33]. Focusing on social bees, we found a stark contrast in the temporal dynamics of insect and plant viruses present in their virome. While host genetic distance strongly shapes the distribution of insect-virus communities, we found plant-pollinator interactions and their phenology to drive plant-virus communities in bee samples. If we consider plant-pollinator

networks as a proxy of virus transmission potential, meta-transcriptomes suggest that not all of this potential is realized as only half of the identified insect viruses were shared between species in this study, indicating some degree of host specificity. In addition, small RNA sequencing revealed that only a subset of successfully transmitted insect viruses appears to replicate in multiple hosts in this assemblage, but among them, we found virulent viruses that drive economically important diseases.

The exchange of pathogens on flowers has been shown to be an important avenue for disease spillover among insects[27,37,38]. However, by measuring plant-pollinator networks, we found that the potential for interspecific virus transmission is not fully realized, as foraging niche overlap does not explain the virome composition of insect viruses in bees. While plant viruses were generally widely spread across pollinators at any one-time point, the presence of insect viruses in meta-transcriptomes varied, with many insect viruses showing a more restricted distribution, with for example Lake Sinai viruses only being found in honeybees. This could indicate that some viruses, such as the *Apis* rhabdoviruses, also not found in taxa other than honeybees, may not be orally transmitted, and thus not spread to new hosts through the shared use of flowers. Other potential explanations include lower stability of particular viruses in the environment or inside potential hosts, as well as differences in tissue tropism that may lead to variation in transmission likelihood. The pathogenicity and the capacity of viruses to reach high loads in their primary hosts, such as DWV and BQCV in honeybees, and SBPV and ABPV in bumblebees[33,38,39], also clearly have the potential to increase cross-species transmission. For instance, viral titer is recognized as a strong determinant of transmission risk in many human diseases[53]. In honeybees, vector-borne transmission by the parasitic mite *V. destructor* is a major factor in increasing viral load[26], and has been shown to increase transmission risks to bumblebees[27,38].

To understand the biological and pathological relevance of the observed viral transmission across bees, we sequenced vsiRNAs and identified viruses showing signs of active in-host replication. Our results revealed that about half of insect viruses identified with meta-transcriptomics were also recognized by the host's RNAi immune system. We found that only a surprisingly small fraction of insect viruses appeared to replicate in multiple host species in this community. This indicates host specificity, mediated through resistance in the host or its microbiome. For instance, only five viruses were identified via vsiRNAs in both honeybees and bumblebees, including ABPV, a virulent virus leading to bee paralysis, and recognized as a serious threat to beekeeping activities[17]. ABPV was also found to be a true multi-host virus at the individual host level in this host assemblage: phylogenetic analysis of individual ABPV sequences as well as the increasing ABPV prevalence and loads observed with time in bumblebees and honeybees showed that this virus circulates freely, without strong barriers between different host species[33].

Two other economically important bee viruses, BQCV and DWV, were also found in many of our transcriptome libraries, and at the individual level by PCR for DWV[33]. However, in contrast to ABPV, we found no siRNA sequences derived from these two viruses in bumblebees, suggesting an absence of replication. Both viruses have been repeatedly detected in wild bumblebees, and several studies demonstrated the potential of BQCV[54–56] and DWV[27,37,54–58] to replicate in naturally infected bumblebees. Recent experimental work, however, reported a low replication rate and limited transmission potential in bumblebees[55,59–61], suggesting lower susceptibility to these two viruses. It should be noted however that, while a vsiRNA signal is diagnostic of viral replication, its absence cannot be categorically interpreted as an absence of replication, as it could simply be below the detection threshold for our pooled samples; and it may also be a less sensitive approach than PCR-based methods such as negative strand detection[27,37]. Some viruses may also evade part of the host antiviral

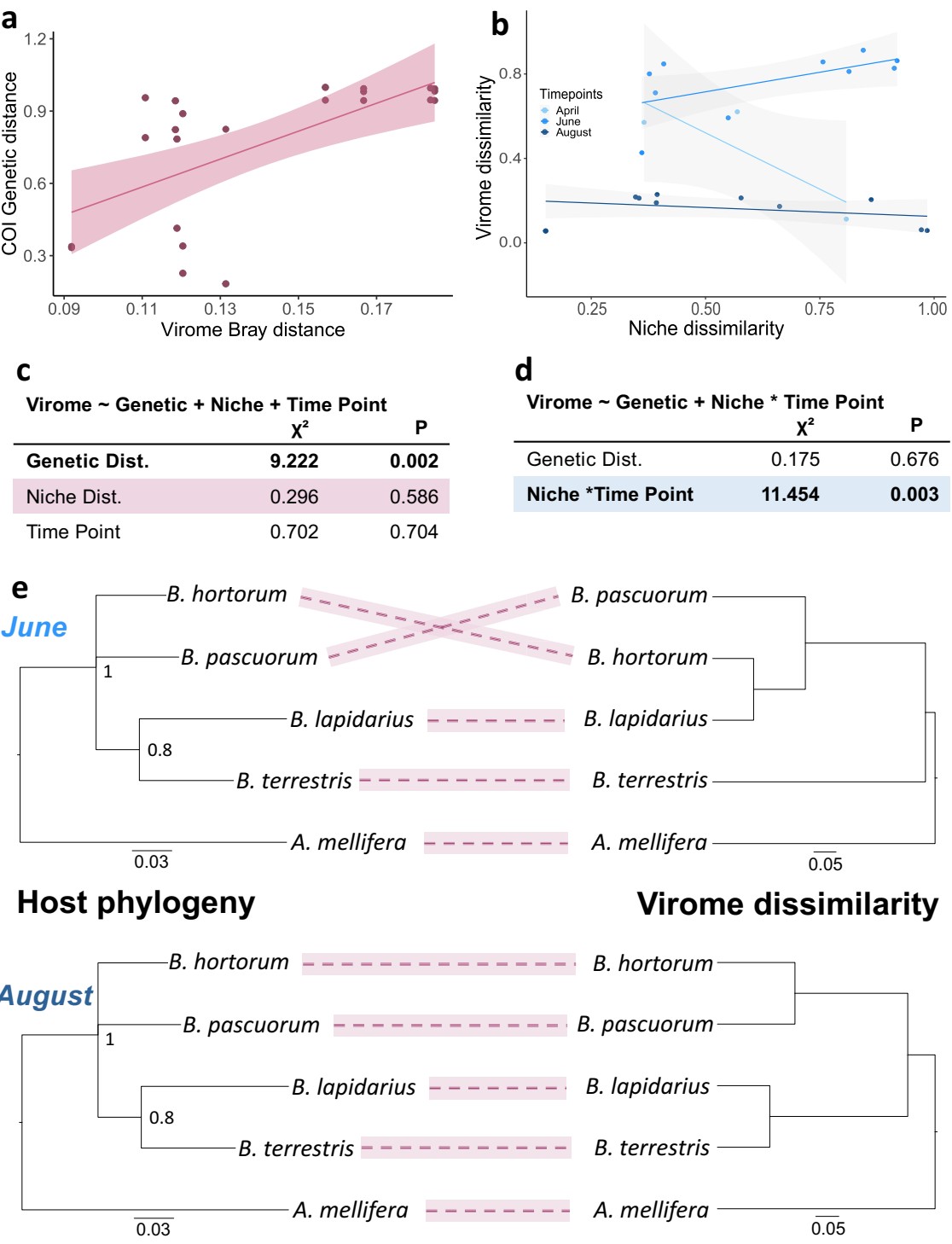

**Fig. 3 | Effect of host phylogeny, host niche, and plant phenology on virome composition. a** Scatter plots showing the effect of host genetic distance on virome composition on insect viruses (Bray-Curtis dissimilarity index), and (**b**) the effects of niche dissimilarity (derived from plant-pollinator networks), by collection time point, on the virome composition in plant viruses (Bray-Curtis index). Plotted lines show the estimated effects, and shaded areas indicate the 95% confidence intervals.

**c, d** The two-tailed Likelihood Ratio Test results from model comparison for insect and plant viruses' assemblages respectively. **e** Co-phylogeny plots generated from COI (left) and virome dissimilarity in insect viruses from June (top) and August (bottom), supporting the effect of host phylogeny on insect virus distribution. Data from April that include only three species were omitted. Source data are provided in Supplementary Data 3.

response, and the reduced number of viruses with detectable vsiRNA signals in multiple bee species could also result from active viral suppressors of RNA interference (VSRs). VSRs can be powerful inhibitors of antiviral defenses, including in insects[62], however, so far no bee virus has been described with VSR.

Overall, these results illustrate how host phylogeny may deeply constrain the capacity of pathogens to infect a new host, and are consistent with previous work showing the effect of host taxonomy on virome composition in mammals[9,13], reptiles[11], birds[10], fish[63], and insects[5,64], including bees[24,44]. The high modularity of host-virus

networks generated here illustrates further the specialization of viromes and is consistent with previous studies[4]. The tight co-evolutionary relationships that enable cell entry and the use of the host cell replication machinery by viruses, before triggering an immune response, are likely to define the viral host spectrum and may have implications for viral ecology and wildlife conservation. For instance, there is a clear link between the current loss of biodiversity and the spread and emergence of infectious diseases. Lower host diversity has been shown to increase viral host jumps in various clades, such as coronaviruses in cave bat populations[65], and has been shown to be the global change driver most strongly associated with an increase in infectious diseases[66]. This is also particularly relevant in the context of pathogen spill-over, which has been well documented from honeybees to wild pollinators[27,37], but for which the consequences in other insects remain poorly understood. The case of DWV, primarily infecting honeybees, but with potentially limited replication potential in bumblebees, is consistent with the dilution effect observed in areas with high pollinator diversity[33]. Increasing bee diversity may thus provide a fragmented host landscape to those viruses that do not replicate equally well in all host species, potentially limiting their spread through dead-end spillover[67]. It remains important to note, however, that viruses may also replicate in phylogenetically distant species like honeybees and bumblebees, as for example seen here for ABPV, a particularly virulent viral species. Indeed, at the individual level, we previously found evidence in this population for a dilution effect for DWV, but not for ABPV, where prevalence was driven by the abundance of its key host species instead[33].

In addition to host phylogeny, other virome studies highlighted the role of ecological interactions across trophic levels, such as host-parasite[64] or predator-prey[4] relationships, on the spread and cross-species transmission of viruses. Here, we found that plant-pollinator networks and their phenology influenced the dynamics of plant viruses carried by bees, and potentially the spread of plant diseases. Our results demonstrated that bee-associated plant virus richness and diversity varied dramatically within a year. The observed decrease in pollinator virome dissimilarity for plant viruses from June to August in our samples may reflect the seasonality in flower provision in agricultural landscapes in temperate regions, which peaks in early summer and decreases towards the end of summer, with August being reported as the most challenging month for insect pollinators[68], leading to changes in plant-pollinator interaction diversity[69]. The tight clustering of our April samples for bee-associated plant viruses is also consistent with the reduction of flower provision in spring, which leads to an increase in bees' foraging niche overlap observed from plant-pollinator networks collected simultaneously[69], providing disease transmission opportunities early in the season. The detection of plant vsiRNAs in our study suggests that plant viruses carried by bees are likely to retain their infectious potential. This may represent a risk of cross-contamination between crop, wild, and ornamental plants by generalist viruses such as ArMV[70], a virus found in almost all our libraries. Plant viruses are often pollen-borne and can in principle be vectored by bees[71], potentially harming wild and cultivated plants[72]. Vectoring of plant diseases by pollinators is a real concern[73], and understanding the combined impacts of insect pollinators and flower seed mixtures in agricultural landscapes on plant virus epidemiology may provide new opportunities for disease spread mitigation.

With this study, we tested the respective roles of host phylogeny and host ecology on the temporal dynamics of bee viromes. We demonstrate that, despite the high potential for interspecific transmission revealed by an overlap in foraging niche and the observed distribution of plant viruses, only a limited fraction of insect viruses are shared across social bee species. Within these shared insect viruses, even fewer are recognized by the RNAi immune system of bees, suggesting that only a limited number of viruses may be truly acting as multi-host pathogens in wild bee communities. Nevertheless, some of these multi-host viruses are virulent pathogens causing significant pathogenicity to bumblebees and damage to honeybees and beekeeping. We found that these important viruses are freely circulating across bumblebee assemblages, raising concerns once more about the potential of disease spillover, particularly from commercial bumblebee colonies used for crop pollination[74]. The identification and quantification of plant viruses in insect meta-transcriptomes illustrates how insect pollinators could be instrumental in monitoring plant diseases for crop and wild plant health[75,76] or act as proxies for plant-pollinator networks, e.g., for canopy foragers in tropical regions, which are inaccessible to behavioral observations. Finally, our results highlight the vectoring capacity of social bees for plant viral diseases within agricultural landscapes, which calls for comprehensive studies of this mechanism and its potential applications.

## Methods

### Sample collection
Insects were collected from farmland with the permission and collaboration of the respective farmers and landowners. Sampling was performed across South East England (in Oxfordshire, Hampshire, and West Sussex counties), at ten farms with different levels of pollinator conservation program implementation, providing a wide variety of flower resources across space and time, including wildflower strips for pollinators along field margins[69]. For this study, we collected insects and plant visitation data from insects (see below) for each farm at three time points across the pollinator season in 2016: in spring (from 19th March to 9th May; referred to as "April"), early summer (from 18th to 30th June, referred to as "June") and late summer (from 30th July to 10th August, referred to as "August"). For each sampling visit, we collected on average 30 of the five most common insect flower visitors based on morpho-groups. Samples were immediately deep frozen in a dry shipper for RNA sequencing and virus analysis[33]. Honeybees and bumblebees, which were nearly always amongst the most common species, were identified to species level. We differentiated between the bumblebee species pairs that are challenging to distinguish in the field (*Bombus terrestris/lucorum* and *Bombus hortorum/ruderatus*) in the laboratory using mitochondrial DNA length polymorphisms before proceeding to RNA extraction (see Supplementary Method 1). Because of their relative scarcity, mining bees, as well as hoverflies and other "Forgotten" flies were considered as morpho-groups for this sampling scheme. See Supplementary Data 1 as well as ref. 33 for details on samples.

### Plant visitation and pollinator networks
Plant-pollinator networks were recorded along transects at each farm site and time point. Transects of 100 m length and 2 m width were selected based on the abundance and richness of flowers and insect visitors within the farm[33,69]. Insect interactions with flowers were recorded by walking along the transects for 15 min. Transects were only performed in favorable conditions, including wind at a maximum of 5 on the Beaufort scale and a minimum shade temperature of 15 °C in summer and 9 °C in spring. Honeybees and bumblebees were identified as species, with the exceptions of the species complexes *B. terrestris/lucorum* and *B. hortorum/ruderatus*, neither of which have workers that are identifiable on the wing.

### RNA library preparation and sequencing
RNA was extracted from laterally bisected, non-surface sterilized bee and fly individuals (except for smaller species such as flies, where whole individuals were used) using a Trizol©/ bromo-chloropropane extraction following homogenization (Invitrogen, Carlsbad, CA, USA). After measuring RNA quantities, samples were pooled in equimolar amounts by species or morpho-groups and time points (Supplementary Data 1). Samples were treated with DNase I and oligo-dT selected to reduce contamination with bacterial ribosomal RNA and increase

the proportions of reads from poly-A-tailed RNA viruses. Library quality was checked using an Agilent 2100 Bioanalyzer before sequencing 100 bp paired-end using Illumina HiSeqTM4000 (BGI Bioinformatics, China). Sixteen libraries of pooled RNA were sequenced generating on average 112 M of paired reads per library (Supplementary Data 1).

## RNA virus characterization and discovery

Paired-end Illumina reads were analyzed for virus discovery following standard methods. After checking the read quality with FastQC[77], adapter sequences and low-quality reads were removed using Sickle[78]. Before assembling raw reads, host sequences were filtered out by mapping against the host genomes using Bowtie2[79] for *A. mellifera* (GenBank GCA_000002195.1) and *B. terrestris* (GenBank GCA_000214255.1). Reads were then assembled de novo using Trinity[80]. We retained all scaffolds with a length of at least 500 nt and grouped the resulting scaffolds into clusters meeting at least a 90% sequence identity threshold using the blastn function from the BLAST+ program[81]. Contigs were then translated over three reading frames and two strains. ORFs from the same contig were concatenated, and we retained only those with an ORF of 150 codons or greater, as in similar studies[24]. These concatenated protein sequences were used to search against a custom database using blastx, retaining a single top hit per contig with an e-value threshold of 0.001. Our custom target database comprised all viral proteins from the Genbank non-redundant protein database and all the hymenopteran and dipteran sequences from NCBI refseq protein database downloaded on February 22, 2021. This reference of viral sequences was used as a target for viral quantification by mapping reads using CoverM (https://github.com/wwood/CoverM) and the bwa-mem method. Each virus was considered to be present in a library if the number of reads was above a conservative threshold of 50, with a minimum coverage threshold of 5% and 250 nt of the target sequence. We report viral abundance after read number normalization by the total number of reads from the library and the length of each target sequence. We grouped putative virus fragments taxonomically according to their initial best blast hit, and manually curated them with reference to closest relatives in GenBank, to identify host taxonomic groups (insect or plant). We removed viral assemblies with unresolved taxonomy (i.e., sequence homology to uncharacterized virus families, $N = 11$) from analyses. A virus OTU was considered novel if it shared <90% amino acid identity with known viruses in our database.

## Small RNA sequencing and small interfering RNA profiles

As the presence of reads from RNA viruses in transcriptomes is not a genuine proof of infection, we sequenced small RNAs to identify viruses triggering an immune response of the host. Upon entry in host cells, replicating viruses trigger an RNAi response, in which virus-derived double-stranded RNA is detected by the Dicer-like proteins and sliced into small RNA molecules for sequence-specific degradation[82]. Insects typically produce sense and antisense vsiRNAs of 21–23 nt long fragments[83]. Bees produce vsiRNA profiles with a peak of 22 nt-long fragments[48], while plants generate peaks at different sizes (21 and 22 nt)[49,50]. Small RNA sequencing was performed from the same pools of RNA used for meta-transcriptome sequencing. Small (18–30 nt long) RNA fragments were separated from longer RNA molecules using a PAGE gel before sequencing with Illumina HiSeq technology (BGI Bioinformatics, China). Sixteen libraries of pooled samples RNA were sequenced generating on average 39 M small reads per library (Supplementary Data 1). A positive siRNA response against viruses was determined with a minimum of 10 reads and a minimum coverage threshold of 5% of the target viral sequence and was used to identify replicating viruses. After mapping small RNAs to viral sequences with CoverM (as for transcriptomes, see above), vsiRNA coverage maps were generated using Samtools depth function from

sorted bam files to count read depth at every position (Supplementary Fig. 2).

## Statistical analyses

Virome composition across our 16 libraries was examined using R. We calculated species richness and Shannon diversity from normalized read counts and tested the effect of taxonomy and collection time points by computing Bray-Curtis dissimilarity as distance measure and performed a PERMANOVA using the adonis2 function of the R package vegan from log transformed normalized read counts[84]. NMDS plots were performed using the metaMDS function, and heatmaps were drawn with two-way hierarchical cluster analysis using the R package pheatmap[85]. Weighted host-virus networks were constructed from normalized counts and analyzed for modularity ($Q$) using the bipartite R package[86]. Modules are formed when nodes have more interactions within the module than among modules, and thus modularity is the result of some degree of specialization in interactions. Modularity $Q$ ranges from 0 for randomly configured networks to 1 for networks composed of perfect modules. We tested network modularity by comparing the observed network $Q$ value against the values of 100 randomly generated networks using the vaznull method that keeps connectance equal to the observed network. We then standardized $Q$ values into z-scores to assess the significance of the observed values (i.e., z-scores > 1.96 are considered significant). Host genetic distance matrix was computed using DNADist on an alignment of a portion of the cytochrome oxidase 1 (CO1) mitochondrial gene available on GenBank: AY181169 for *B. terrestris*, AY181102 for *B. hortorum*, AY181114 for *B. lapidarius*, KR005519 for *B. pascuorum* and NC_051932 for *A. mellifera* (Supplementary Data 3). We calculated species richness, alpha (Shannon index), and beta diversity (Bray-Curtis) from normalized viral read counts and measured pairwise foraging niche dissimilarity (Horn-Morisita) from the plant-pollinator networks[87] using the R package vegan[84]. Virome beta diversity measures were analyzed with linear mixed models using the R packages lme4[88] and blme[89], using host genetic distances and the interacting factors niche overlap and collection time point as fixed effects, and host pairs as random effects. Single-term significance was assessed using likelihood ratio tests. All models were checked for overdispersion using the overdisp_fun function.

## Reporting summary

Further information on research design is available in the Nature Portfolio Reporting Summary linked to this article.

## Data availability

Meta-transcriptomes and small RNA sequences generated in this study are available in the NCBI SRA database under the BioProject PRJNA1110080. Plant pollinator interaction data are available at https://doi.org/10.5061/DRYAD.MSBCC2G2Q[87]. Source Data for Figs. 1, 2, and 3 can be found in Source Data file and Supplementary Data 2 and 3. Viral assemblies and mapping outputs used for Supplementary Figs. 2 and 3 are available at https://doi.org/10.6084/m9.figshare.27888378. Source data are provided with this paper.

## Code availability

R scripts are available at https://doi.org/10.6084/m9.figshare.27888378

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

## Acknowledgements

We are very grateful to all of the farmers involved—this work would not have been possible without their generous support in giving us access to their land. We would like to thank Emily Bailes, Charlotte Stewart, Isobel Refoy, and Sophie Hedges for help with collecting samples and data in the field, Jess Lewis for help with processing samples in the lab, Carina

Scheifele for suggestions on graphical representation, and Jana Dobelmann for comments on the manuscript and data analyses. We also thank anonymous reviewers for their constructive comments on an earlier version of this manuscript. The authors acknowledge support by the High Performance and Cloud Computing Group at the Zentrum für Datenverarbeitung of the University of Tübingen for the use of bwForCluster BinAC cluster facility, the state of Baden-Württemberg through bwHPC and the German Research Foundation (DFG) through grant no. INST 37/935-1 FUGG. The contribution of C.C. was supported by the Natural Environment Research Council (NERC) under research program NE/N018125/1 ASSIST–Achieving Sustainable Agricultural Systems www.assist.ceh.ac.uk. ASSIST is an initiative jointly supported by NERC and BBSRC. This work was supported by the BBSRC (BB/N000625/1 and BB/N000668/1) to L.W. and M.J.F.B.

## Author contributions

Conceptualization: V.D., C.C., M.J.F.B., L.W. Funding acquisition: M.J.F.B., L.W. Resources: L.W., C.C. Investigation: V.D., T.D. Data analysis: V.D. Visualization: V.D. Project administration: L.W. Writing–original draft: V.D. Writing—review and editing: V.D., T.D., C.C., M.J.F.B., L.W.

## Funding

## Competing interests

The authors declare no competing interests.
