## [Transparent Peer Review file · Nature Communications]

Host ecology and phylogeny shape the temporal dynamics of social bee viromes

Corresponding Author: Vincent Doublet

Version 0:

Reviewer comments:

Reviewer #1

(Remarks to the Author)

Summary

This is a well-written and concise manuscript that packs a punch with novel and interesting data. The observational study design is similar to a few other studies that have been published recently (including previous studies by the authors), but in my opinion the combination of transcriptomic and siRNA approaches, in addition to host phylogenetic and network analyses, facilitates inference that really pushes forward knowledge on the topic of multi-host pathogen sharing among pollinators. I haven't seen anything like this manuscript in the literature; I enjoyed reading it and learned several things.

I don't have any major critiques, but I do have several minor comments. First, it's a bit unclear when the data from non-social pollinators are included in analyses and when they aren't included. The metatranscriptome summary in Fig 1 shows the mining bees and flies, then the mining bees and flies are missing from Fig 2, and I assume they're also missing from Fig 3. I don't understand the rationale for omitting the mining bees and flies from Fig 2 (and also omitting this info from the main text and supplement). Perhaps siRNA analyses weren't conducted on these groups? I think I understand the rationale for omitting them from Fig 3 (pooled samples, very distant phylogenetically compared to Apidae), but this rationale is never presented as far as I can tell. Including this info in the methods or briefly in the results narrative could avoid confusion. I'm curious if SBPV and other viruses are replicating in *Andrena*, etc!

Second, isn't the top network in Fig 2b identical to the info conveyed in Fig 1b? It's clear that honey bee viromes (black) are fairly different than bumble bee viromes (browns and oranges), but we already know that from Fig 1b. Perhaps cut that network? Conversely, the siRNA network is very compelling in showing that honey bees are unique from bumble bees (and bumble bees have a lot of overlap). This said, I really want to know the identity of all the siRNAs (shared and not) in the bottom panel. Fig 2a doesn't provide the detail necessary for this. Could a table (or binary pres/absence map?) be used to show this info in either the main text or supplemental? These are potentially very interesting details that aren't currently included in the manuscript.

Title

What are the spatial dynamics that are shaped by host ecology and phylogeny? Maybe drop "spatio"?

Abstract

Lines 17 and 19: Perhaps add parentheses around the examples.

Introduction

Lines 47-61: The exclusive focus on viruses in this paragraph misses all of the non-virus pollinator EEID literature on this topic. That's probably fine since the major concepts are conveyed, but there are several highly relevant papers that aren't referenced here as a result (e.g., Graystock et al. 2020 *Nature Ecol & Evol*).

Line 60: I recently had a lengthy discussion with four systematics experts, three honey bee researchers, and a journal editor and we agreed it's most appropriate to use a lowercase "w" for the "western honey bee". This is a common name for a

species, which is lowercase by convention unless it's a proper noun. "Western" was never meant to stipulate the species only inhabits the Western Hemisphere (it doesn't, even for historically natural populations) or any specific geographic location. Instead, it's a general reference to geography (i.e., eastern versus western), so it isn't a proper noun and shouldn't be capitalized. But "Western" and "western" are both in the literature in approximately equal amounts. All of us agreed to start being consistent and use "western honey bee" or "western honeybee" from here forward. Here I'm attempting to spread the word...

Results

Line 103: Write out "sacbrood virus (SBV)".

Line 109: "but all three share little of their virome with bumble bees and honey bees." I'm not sure I agree with this statement – there's actually a decent amount of overlap.

Figure 1: Really nice figure, great job. Can labels (and units) be added to the pink and green scale bars so we know what the intensity of shading represents?

Lines 133-135: What about siRNAs in the mining bees and flies?

Lines 141-142: Definitely a surprising result given the other literature on this topic from this research group. I was surprised to see somewhat limited discussion of this result (line 225-233). Could there be methodological reasons for limited evidence for BQCV and DWV replication in bumble bees? Or is this a real result? If it is real, it seems to suggest disease spillover isn't likely. A bit more discussion on this topic would be appreciated.

Line 164: These statistics aren't present in Fig 2c. It looks like the relevant stats are on line 170?

Line 170: I think these statistics are supposed to be linked to Fig 2c, not Fig 2d. Right?

Lines 171-172: Is the June result statistically significant?

Figure 3e: What do the host phylogeny – virome dissimilarity associations look like for April and August? Similar to June?

Discussion

Lines 251-261: This is a rather limited discussion of the plant virus results. I'm curious why different time points had different relationships (Fig 3b), and why virome dissimilarity was much higher early in the summer compared to late in the summer. Those key results aren't really discussed here.

Lines 258-259: There are many ecologically and economically important plant diseases that are vectored by pollinators. A brief summary is contained in McArt et al. 2014 Ecology Letters.

Methods

Line 306: Were the insects surface sterilized? If not, what is the likelihood some of the viruses (and siRNAs) were on the exterior of the insects instead of interior? How might that influence inferences regarding the results?

Reviewer #2

(Remarks to the Author)

Comments to authors

The authors used a total RNA sequencing and small RNA sequencing approach to study the effect of host genetic distance on bee virus distribution, and how plant-pollinator interactions and phenology influence plant virus communities. The study is well designed, well written and with good quality figures. However, I have concerns at the lack of statistical analysis to back up statements made in the results, and the lack of discussion of previous research and how their research fits into the broad knowledge of virus evolution and ecology, beyond other studies of bees/plant-pollinator systems. I'm also concerned about how the plant/insect viruses were distinguished, and why some viruses were detected in vsRNA signals but not the metatranscriptomic data. More details below.

- Page 2 line 35. Symbiosis usually refers to a mutually beneficial relationship so the term 'pathogenic symbiont' does not sound right.
- Page 2 line 43 – 45. Yes rarely, but it has been explored. You still need to acknowledge the research that has been conducted in this space, and the implications of this for your research. This applies to both the introduction and the discussion.
- Page 3 line 74 – 75. You need to include Latin names even if these are species groups. E.g. hoverflies are presumably Syrphidae spp.?
- In total RNA sequencing data there are often hits to divergent viruses e.g. 'Riboviria sp'. What did you do with these viruses? Did you exclude them or classify them as insect or plant virus?
- Page 3 line 90. So were all vsRNA signals that were 21nt fragments assumed to be plant viruses?
- Page 3 line 100 – 101. Often you state a result and link to a figure, but there doesn't appear to be any statistics to back this up. For this sentence 'viromes restricted to insect viruses clustered mainly by host taxa' you could confirm this using a PERMANOVA analysis to determine if the viromes of different species were significantly different from one another. If you have actually done statistics on this, it needs to be made clearer. There are many other statements like this in the results

where they don't appear to be backed up by statistics.

- Page 5 line 129 – 132. Not convinced about this. In reference 38 the viruses they found infecting the bees had a distribution of 21-22nt long, not 22 only. A significant chunk of their vsiRNA for the bee viruses were 21 nt long. In ref 39 I can't find anything saying the plant RNAs are 21 nt long, but in another paper (<https://www.mdpi.com/2073-4425/8/6/163>) you can see there also some plant vsiRNA that is 21 nt long. It isn't clear how exactly you distinguished between the plant and insect viruses, but I think you would need to look at the distribution of the vsiRNA sequence lengths rather than a strict 21/22nt cut-off.

Also, Would these vsiRNA signals get degraded quicker than the virus genome? Is it possible that you are missing signals from infecting viruses because of this?

- Page 5 line 136. In the venn diagram I count 6 viruses that were found in vsiRNA signals, not 24. I think you've included ones found using vsiRNA in one species and the transcriptome in another which is not valid if you're assuming viruses are only multi-host if they are infecting both host species.

- Figure 2A. It looks like in the *Bombus* spp. you have 7 viruses that were only detected via vsiRNA. How do you explain this?

- Figure 2B. This would be strengthened by network analysis (rather than just creating the network) e.g. community detection algorithms.

- Page 8. Discussion needs broadening - wider implications of our understanding of virus evolution, cross-species transmission, and comparison to other studies beyond studies of bees/plant-pollinator systems.

- Page 12 line 328. When was the genbank protein database downloaded?

- Page 12 line 333. 50 reads seems quite high. Maybe this is why you detected some vsiRNA signals that you didn't detect in your transcriptome data? Have you tried looking for these missing viruses to see if they are present at a lower abundance? If so, assuming the vsiRNA signal proves infection, it would suggest the 50 read limit is too high.

Reviewer #3

(Remarks to the Author)
Editors,

The manuscript entitled, "Host ecology and phylogeny shape the spatio-temporal dynamics of social bee viromes" by Doublet et. al Wilfert describes viral transcripts/RNA genomes and vsiRNAs in RNA sequencing libraries obtained from social bees honey bees, and four bumble bee species, one solitary bee species (*Andrena* spp) and two flies (i.e., Hover flies and forgotten flies). The manuscript highlights some interesting and important findings including that of the ~24 insect viruses detected in multiple bee hosts in this study, only a five actively replicating viruses are likely shared between honey bees and bumble bees (i.e. ABPV, Castleton Burn virus, Phle1, Hubei partiti-like virus 34, and anew Osugoroshi-like virus) (i.e., based on vsiRNA sequences).

It seems there is much more information that could be revealed by this dataset including, the full genome sequences and coverage maps of the main viruses detected and describe in this study, as well as the potential site and/or site/time specific differences in the viromes present in different bee species at the different sites, or at different sites and times.

Points to clarify or address before publication include:

1. Abstract Line 25 – the word "sequences" should be added to the phrase "insect and plant virus sequences".

2. Abstract Line 25 – The text states that 6,000 insect samples were obtained, but the description of these samples is incomplete – and it seems the RNA from only a subset of these samples was pooled into 16 sequencing libraries (Supplemental Table S1). The sample list should include more details (e.g. geographic location/site, exact sample date, etc.). The authors should include more details about the number of each species at each site (i.e., 10 sites, x8 species, x 3 time points) that make up the pooled samples sequenced for this study. Another supplemental table should be included.

3. Abstract Line 28 – the word "diseases" should be replaced with "viruses", or perhaps both words can be used.

4. Lines 208, 210 and throughout the manuscript "virus titer" should be changed to "virus RNA copy number", as "titer" is typically used to indicate virion number as assessed by plaque assays and the data presented in this paper is based on RNA sequence abundance (i.e., virus RNA copy number per xxxx ng RNA), which includes both genomic RNA and transcripts. Relative virus abundance may be ok too.

5. Lines 196-197 –

Consider using "transmission" or "spread" rather than spillover

The authors should use "transmission" rather than "spillover/spillover" since many studies have shown that BQCV (and DWV) are prevalent in both honey bees and bumble bees, so these viruses are likely just shared between bee species within a community. Viruses are transmitted between different genera / species of bees – so that use of the word "spillover" is not always appropriate. Although, there may be several definitions of the word "spillover" (e.g., Wikipedia "Spillover infection, also known as pathogen spillover and spillover event, occurs when a reservoir population with a high pathogen prevalence comes into contact with a novel host population. The pathogen is transmitted from the reservoir population and may or may not be transmitted within the host population.") doesn't seem accurate for bee viruses. Additional temporal studies are required to determine the ecology of bee viruses – as well as to describe these events as either potential spillover or spill back.

6. Line 224 – “disease” should be replaced with the word “virus” – since DWV is a virus, not a disease.

7. For the main viruses discussed (DWV-A, DWV-B, ABPV, SBPV, Castleburn virus, and others), the authors should map their sequencing date back to the viral genomes and describe the depth of sequence coverage, overall genome coverage, percent identity, etc. This data should be included as figures or tables (maybe one main figure, and supplemental info). The virus genome sequencing data for all of the described viruses should be deposited on NCBI and accession numbers added to the text. The percentage identity shared with the most similar genome on NCBI should also be listed.

8. The authors could include more detail and description about the sequences in each library (e.g., were the five many viruses detected in all libraries generated from all sample dates, were all of those viruses detected at all 10 sites (if samples were tagged by site) The methods section indicates 17 libraries were sequenced by the text states 16. In addition, to the number of total reads obtained for each sequencing library, the authors should include more analyses information in the Supplemental Table S1 (e.g., the number/percentage of reads that aligned to viruses in each library, the reads assigned to specific key viruses in each library, etc.). The numbers should have “,” rather than decimal places. It would be great if the authors could share more of their sequence data in an analyzed format (e.g., virus sequences binned by virus, etc.).

9. Depending on how much of the Castleburn virus the authors assembled (e.g., only RdRp sequence vs. entire genome), it may be that they are detecting a recently described virus (i.e., Andrena associated bee-virus 1 (AnBV1). AnBV-1 has a bipartite RNA genome (RNA (MW397641.1 – 2,721 bp with RdRp and MW397640.1 RNA, 2005 bp) with the RdRp containing RNA sharing regions of high identity to Castleburn virus (GenBank: MH614293.1 2,714 bp)(BLAST nucleotide alignment $4e-30$ 246/363 - 68% identity).

see Daughenbaugh et al, *Viruses* 2021, 13(2), 291; <https://doi.org/10.3390/v13020291>

“The putative 499 aa AnBV-1 RdRp sequence produces the strongest alignment with a putative Castleton Burn virus RNA-dependent RNA polymerase, (BLASTp e-value = 0), which was identified by sequencing bumble bee samples (Supplementary Figure S11) [156]. The RNA-dependent RNA polymerase proteins of AnBV-1 and Castleton Burn virus are similar. However, they share only 53.3% amino acid identity (Supplementary Figure S11).”

10. Figure 2A – should include a table with the shared virus names (~ 36 viruses), the 2, 2, and 6 shared viruses could be listed on the figure, and the others should be included in a supplemental table.

11. Line 163 “pollinator-associated or bee-associated” should be added before the words “plant viruses” to clarify the meaning of this sentence.

12. Line 191, should be rephrased. This study involved sequencing of insect associated viruses at three distinct times in Scotland, and the data included are insufficient to state that “many insect viruses not able to jump from one host to another”. It would be better to state that “not many insect viruses (or virus sequences) were shared between species in this study, which may indicate host specificity”.

13. The vsiRNAs should be mapped back to select virus genomes to illustrate virus genome coverage, and provide additional support for replicating viruses. It is great that the authors describe this data, but the figures do not well-represent these findings.

14. Line 125 – change the word “pathogens” of “multi-host pathogens” since this study only demonstrates that these are bee-associated virus sequences, it would be better to just call them “viruses” – since the authors did not demonstrate pathogenicity.

Minor points to clarify or address before publication include:

1. Key words “honey bee” and “bumble bee” are two words, whereas they are written as compound words throughout the manuscript.

Version 1:

Reviewer comments:

Reviewer #1

(Remarks to the Author)

I think the authors have done an excellent job with the revision. All of my comments have been addressed with additional analyses and/or text that provides further clarity, context, and support of manuscript's main conclusions. My feeling is that comments from other reviewers have also been addressed satisfactorily, but I'll defer to their opinions and expertise. Thank you for a very nice research effort, I look forward to citing this paper.

Reviewer #2

(Remarks to the Author)

All my concerns and comments have been adequately addressed, thank you for your responses.

Reviewer #3

(Remarks to the Author)

Reviewer comments, and authors' response (*in blue*)

Reviewer #1 (Remarks to the Author):

Summary

This is a well-written and concise manuscript that packs a punch with novel and interesting data. The observational study design is similar to a few other studies that have been published recently (including previous studies by the authors), but in my opinion the combination of transcriptomic and siRNA approaches, in addition to host phylogenetic and network analyses, facilitates inference that really pushes forward knowledge on the topic of multi-host pathogen sharing among pollinators. I haven't seen anything like this manuscript in the literature; I enjoyed reading it and learned several things.

I don't have any major critiques, but I do have several minor comments. First, it's a bit unclear when the data from non-social pollinators are included in analyses and when they aren't included. The metatranscriptome summary in Fig 1 shows the mining bees and flies, then the mining bees and flies are missing from Fig 2, and I assume they're also missing from Fig 3. I don't understand the rationale for omitting the mining bees and flies from Fig 2 (and also omitting this info from the main text and supplement). Perhaps siRNA analyses weren't conducted on these groups? I think I understand the rationale for omitting them from Fig 3 (pooled samples, very distant phylogenetically compared to Apidae), but this rationale is never presented as far as I can tell. Including this info in the methods or briefly in the results narrative could avoid confusion.

Response: *Thank you for identifying a lack of clarity around the use of samples for our different analyses. Indeed, we excluded mixed-species pools for the analysis of vsiRNAs, as RNAi response from a pool of species does not make much sense, and for the models with both phylogenetic and ecological distances as explanatory variables (measured from COI gene sequence and plant-pollinator networks, respectively) that require species level information. Therefore, we retained single-species pool viromes from *Apis mellifera*, *Bombus terrestris*, *B. lapidarius*, *B. hortorum* and *B. pascuorum*, and excluded the other datasets. We have now added the following sentence (now lines 194-197) at the beginning of the result section in question:*

*“Using meta-transcriptomes from single-species pools only (i.e. discarding species pools for *Andrena* spp., hoverflies and the ‘Forgotten flies’), we examined the effects of host phylogenetic distance, host ecological distance and collection time points on the beta-diversity of social bee viromes, and found contrasting effects for insect and bee-associated plant viruses.”*

I'm curious if SBPV and other viruses are replicating in *Andrena*, etc!

Response: *We agree that this basic information was missing from our first submission. We now provide vsiRNA profiles (Supplementary figure S5) and coverage maps of viral sequence targets with small RNA reads (Supplementary figure S4) across our libraries in supplements.*

*To answer the question of what is replicating in mining bees: no, SBPV is not found replicating in our pool of *Andrena* collected over the three collection time points, neither ABPV, BQCV, DWV-A, DWV-B, SBV, ARV1, ARV2, ARV5, CBPV, LSVs, Mayfield virus 1, VOV-1, to refer to the most common bee viruses. However, we found several new viruses*

replicating, and previously described viruses such as Andrena anphe-related virus 1 (described by Daughenbaugh et al. 2021, doi:10.3390/v13020291), Andrena haemorrhoea nege-like virus (Schoonvaere et al. 2018, doi:10.3389/fmicb.2018.00177), Bee Iflavirus 1 (Schoonvaere et al. 2018, doi:10.3389/fmicb.2018.00177) and Castleton Burn virus.

Second, isn't the top network in Fig 2b identical to the info conveyed in Fig 1b? It's clear that honey bee viromes (black) are fairly different than bumble bee viromes (browns and oranges), but we already know that from Fig 1b. Perhaps cut that network? Conversely, the siRNA network is very compelling in showing that honey bees are unique from bumble bees (and bumble bees have a lot of overlap). This said, I really want to know the identity of all the siRNAs (shared and not) in the bottom panel. Fig 2a doesn't provide the detail necessary for this. Could a table (or binary pres/absence map?) be used to show this info in either the main text or supplemental? These are potentially very interesting details that aren't currently included in the manuscript.

Response: *We agree and are now providing a figure (Supplementary figure S4) that displays the list of viruses with vsRNA signals across our libraries in supplement, see comment above.*

Concerning the top network figure (2B), the reviewer is correct that it is generated from the same data as now Figure 1D. Although this may sound redundant, we believe these graphical representations are complementary and should remain in the manuscript. With Figure 1, we analysed the composition of viromes to identify the samples showing the same viral assemblage across all pools. In Figure 2, we compare meta-transcriptomes and vsRNA sequencing in two networks to show not only that honeybee viromes are dissimilar from bumblebee viromes, as shown in Figure 1D, but also that network modularity remains when we look at replicating viruses only. This latter point is illustrated by the superimposition of the two networks in Figure 2B.

Title

What are the spatial dynamics that are shaped by host ecology and phylogeny? Maybe drop "spatio"?

Response: *We used this term to refer to the spatial dimension of the evolutionary landscape in which the viruses evolve (here in fact the host species); in hindsight, we agree that this is confusing and have removed it from the title.*

Abstract

Lines 17 and 19: Perhaps add parentheses around the examples.

Response: *We realize that these sentences were unclear, and added parentheses line 17, and changed 'e.g.' by 'potentially' in line 19.*

Introduction

Lines 47-61: The exclusive focus on viruses in this paragraph misses all of the non-virus pollinator EEID literature on this topic. That's probably fine since the major concepts are conveyed, but there are several highly relevant papers that aren't referenced here as a result (e.g., Graystock et al. 2020 Nature Ecol & Evol).

Response: *Agreed. This section absolutely benefits by broadening the literature review to other bee diseases. We now refer in line 55 to microparasites and cite the following papers:*

#22 Van Wyk et al. (2021) *Big bees spread disease: body size mediates transmission of a bumble bee pathogen. Ecology 102, e03429.*

#31 Bartlett et al. (2019) *Industrial bees: The impact of apicultural intensification on local disease prevalence. Journal of Applied Ecology 56, 2195–2205.*

#34 Graystock et al. (2020) *Dominant bee species and floral abundance drive parasite temporal dynamics in plant-pollinator communities. Nature Ecology & Evolution 4, 1358–1367.*

#35 Adler et al. (2018) *Disease where you dine: plant species and floral traits associated with pathogen transmission in bumble bees. Ecology 99, 2535–2545.*

#36 Figueroa et al. (2020) *Landscape simplification shapes pathogen prevalence in plant-pollinator networks. Ecology Letters 23, 1212–1222.*

Line 60: I recently had a lengthy discussion with four systematics experts, three honey bee researchers, and a journal editor and we agreed it's most appropriate to use a lowercase "w" for the "western honey bee". This is a common name for a species, which is lowercase by convention unless it's a proper noun. "Western" was never meant to stipulate the species only inhabits the Western Hemisphere (it doesn't, even for historically natural populations) or any specific geographic location. Instead, it's a general reference to geography (i.e., eastern versus western), so it isn't a proper noun and shouldn't be capitalized. But "Western" and "western" are both in the literature in approximately equal amounts. All of us agreed to start being consistent and use "western honey bee" or "western honeybee" from here forward. Here I'm attempting to spread the word...

Response: *Thank you for this interesting comment. We changed it to 'western' (now line 68).*

Results

Line 103: Write out "sacbrood virus (SBV)".

Response: *Thank you for pointing that out. We realized that both deformed wing virus and sacbrood virus were introduced earlier in the manuscript, and added their abbreviation there (lines 67-68).*

Line 109: "but all three share little of their virome with bumble bees and honey bees." I'm not sure I agree with this statement – there's actually a decent amount of overlap.

Response: *Thanks for pointing this out, there is indeed a fair number of viral species shared between this group and the social bees. We changed this to accurately reflect that viromes from mining bees and flies show "a smaller overlap with viromes from bumblebees and honeybees" (now line 129).*

Figure 1: Really nice figure, great job. Can labels (and units) be added to the pink and green scale bars so we know what the intensity of shading represents?

Response: *Thanks for spotting the missing labels and units, we have added them.*

Lines 133-135: What about siRNAs in the mining bees and flies?

Response: *As mentioned above, we are now providing two supplementary materials (S4 and S5) that display the list of viruses showing vsiRNA signal in all libraries, with read coverage maps and vsiRNA profiles. Note that for the two dipteran pools, our small RNA libraries were almost completely dominated by host sequences with less than 1% of reads mapping to our list of assembled viral sequences.*

Lines 141-142: Definitely a surprising result given the other literature on this topic from this research group. I was surprised to see somewhat limited discussion of this result (line 225-233). Could there be methodological reasons for limited evidence for BQCV and DWV replication in bumble bees? Or is this a real result? If it is real, it seems to suggest disease spillover isn't likely. A bit more discussion on this topic would be appreciated.

Response: *Thank you for this comment. Indeed, the absence of vsiRNA signal for BQCV and DWV in bumblebees is intriguing and deserves more attention. It is difficult to tell whether this truly reflects an absence of replication or is due to detection limits. Both viruses were detected in all bumblebee transcriptomes, but at relatively low levels. It is possible that too few individuals were positive to be able to detect vsiRNAs. Both viruses have been repeatedly found in wild bumblebees, and several studies showed active replication of BQCV and DWV with negative strand PCR assays (references in the manuscript). However, recent experimental work also showed limited replication and transmission rates for these viruses in bumblebees, in comparison to their performance in the honeybee *Apis mellifera*, suggesting that bumblebees may be more resistant.*

We have added all these elements in our discussion which we believe will stimulate more research to understand the relationship between bumblebees and these viruses, to understand the role of bumblebees in the ecology and evolution of these important viruses, and the antiviral immune response of bumblebees. This new section of the discussion (lines 261-275) reads as follows:

“Two other economically important bee viruses, BQCV and DWV, were also found in many of our transcriptome libraries, and at the individual level by PCR for DWV³³. However, in contrast to ABPV, we found no siRNA sequences derived from these two viruses in bumblebees, suggesting an absence of replication. Both viruses have been repeatedly detected in wild bumblebees, and several studies demonstrated the potential of BQCV⁵⁴⁻⁵⁶ and DWV^{27,37,54-58} to replicate in naturally infected bumblebees. Recent experimental work, however, reported a low replication rate and limited transmission potential in bumblebees^{55,59-61}, suggesting lower susceptibility to these two viruses. It should be noted however that, while a vsiRNA signal is a diagnostic of viral replication, its absence cannot be categorically interpreted as absence of replication, as it could simply be below the detection threshold for our pooled samples; and it may also be a less sensitive approach than PCR-based methods such as negative strand detection^{27,37}. Some viruses may also evade part of the host antiviral response, and the reduced number of viruses with detectable vsiRNA signals in multiple bee species could also result from active viral suppressors of RNA interference (VSRs). VSRs can be powerful inhibitors of antiviral defenses, including in insects⁶², however, so far no bee virus has been described with VSR.”

Line 164: These statistics aren't present in Fig 2c. It looks like the relevant stats are on line 170? Line 170: I think these statistics are supposed to be linked to Fig 2c, not Fig 2d. Right?

Response: *We are sorry for the confusion about the report of statistical results. In the original manuscript, we showed both model summaries from ANOVAs (panels C and D in Figure 3), and likelihood ratio tests (LRTs) for single parameter tests in the text. As for testing the effect of single parameters, refitting the model by dropping single terms with LRT is a better practice, therefore we changed panels C and D of Figure 3 with results from LRTs. We have added this information in the method (line 447):*

“Single term significance was assessed using likelihood ratio tests.”

Lines 171-172: Is the June result statistically significant?

Response: *Here, the reviewer asks if the observed increased dissimilarity in virome composition in plant viruses between insects with more distant ecological niches (“June” samples in Figure 3B) is significant. After testing it, we found a non-significant relationship ($P = 0.07$) and therefore discarded this sentence.*

Figure 3e: What do the host phylogeny – virome dissimilarity associations look like for April and August? Similar to June?

Response: *This is a very interesting question, for which we thank the reviewer. We have generated the same plot for August and added it to our Figure 3E. Interestingly, it shows a lower discordance (0.33, instead of 1 for June) between the phylogenetic and virome dissimilarity trees. For April, however, it does not really make sense to co-analyse phylogeny vs. virome dissimilarity as presented in Figure 3E, as only three species were sampled in spring, with *Bombus hortorum* and *Bombus pascuorum* still hibernating.*

Discussion

Lines 251-261: This is a rather limited discussion of the plant virus results. I'm curious why different time points had different relationships (Fig 3b), and why virome dissimilarity was much higher early in the summer compared to late in the summer. Those key results aren't really discussed here.

Response: *We are glad for this opportunity to extend our discussion on plant virus dynamics. We now discuss this important result, and the factors that likely influenced the seasonality of bee-associated plant virus richness and diversity. We hypothesise that the key variable influencing the richness and therefore the observed changes in virome dissimilarity across season is flower provision. It has been reported many times, mostly from studies on the honeybee *Apis mellifera*, that the abundance and diversity of flower species that provide nectar and pollen to pollinators varies greatly within a year. Indeed, we also found this seasonality in the plant-pollinator interactions in this field study (Doublet et al 2022, doi:10.1002/ece3.9442), and found that flower diversity and density in agricultural areas influence bee abundance and plant-pollinator network structures. With higher flower provision and diversity in June, we expect to see higher richness and diversity of plant viruses on insect flower visitors.*

This is now discussed from line 300 to line 320, and reads as follows:

“In addition to host phylogeny, other virome studies highlighted the role of ecological interactions across trophic levels, such as host-parasite⁶⁴ or predator-prey⁴

relationships, on the spread and cross-species transmission of viruses. Here, we found that plant-pollinator networks and their phenology influenced the dynamics of plant viruses carried by bees, and potentially the spread of plant diseases. Our results demonstrated that bee-associated plant virus richness and diversity varied dramatically within a year. The observed decrease in pollinator virome dissimilarity for plant viruses from June to August in our samples may reflect the seasonality in flower provision in agricultural landscapes in temperate regions, which peaks in early summer and decreases towards the end of summer, with August being reported as the most challenging month for insect pollinators⁶⁸, leading to changes in plant-pollinator interaction diversity⁶⁹. The tight clustering of our April samples for bee-associated plant viruses is also consistent with the reduction of flower provision in spring, which leads to an increase in bees' foraging niche overlap observed from plant-pollinator networks collected simultaneously⁶⁹, providing disease transmission opportunities early in the season. The detection of plant vsRNAs in our study suggests that plant viruses carried by bees are likely to retain their infectious potential. This may represent a risk of cross-contamination between crop, wild and ornamental plants by generalist viruses such as Arabis mosaic virus (ArMV)⁷⁰, a virus found in almost all our libraries. Plant viruses are often pollen-borne and can in principle be vectored by bees⁷¹, potentially harming wild and cultivated plants⁷². Vectoring of plant diseases by pollinators is a real concern⁷³, and understanding the combined impacts of insect pollinators and flower seed mixtures in agricultural landscapes on plant virus epidemiology may provide new opportunities for disease spread mitigation."

Lines 258-259: There are many ecologically and economically important plant diseases that are vectored by pollinators. A brief summary is contained in McArt et al. 2014 Ecology Letters.

Response: *Thank you for pointing out this reference. We have added it to our discussion (now reference #73, cited line 318).*

Methods

Line 306: Were the insects surface sterilized? If not, what is the likelihood some of the viruses (and siRNAs) were on the exterior of the insects instead of interior? How might that influence inferences regarding the results?

Response: *No, insects were not surface sterilized. Our rationale was to identify the whole RNA virome of pollinators, including plant viruses, to compare the dynamics of plant and insect viruses in the meta-transcriptome, complemented with small RNA sequencing to identify replicating (i.e. infecting) viruses. In consequence, the meta-transcriptomes certainly contain viruses that bees were just in contact to, but likely never infected the bees. This is the case of course for plant viruses, but likely for insect viruses, and may be what we found with e.g. BQCV and DWV in bumblebees (but see discussion in the manuscript for potential limits of detection). In other words, we see our meta-transcriptome as a virus "exposome" study, and the small RNA sequencing as a true virome analysis.*

As for the consequences for the small RNA analysis, we believe that only insect viruses that penetrated host cells, and are replicating, show a vsRNA signal. We also identified replicating plant viruses, but in all likelihood inside pollen grains present on the surface of insect bodies, and not the insect itself.

To clarify the methodology, we have added the phrase “non-surface sterilized” in the method (line 369).

Reviewer #2 (Remarks to the Author):

Comments to authors

The authors used a total RNA sequencing and small RNA sequencing approach to study the effect of host genetic distance on bee virus distribution, and how plant-pollinator interactions and phenology influence plant virus communities. The study is well designed, well written and with good quality figures. However, I have concerns at the lack of statistical analysis to back up statements made in the results, and the lack of discussion of previous research and how their research fits into the broad knowledge of virus evolution and ecology, beyond other studies of bees/plant-pollinator systems. I'm also concerned about how the plant/insect viruses were distinguished, and why some viruses were detected in vsRNA signals but not the metatranscriptomic data. More details below.

- Page 2 line 35. Symbiosis usually refers to a mutually beneficial relationship so the term 'pathogenic symbiont' does not sound right.

Response: We changed 'symbionts' for 'microbes' to avoid this confusion (now line 36).

- Page 2 line 43 – 45. Yes rarely, but it has been explored. You still need to acknowledge the research that has been conducted in this space, and the implications of this for your research. This applies to both the introduction and the discussion.

Response: Thank you for this suggestion. We gladly added more background in the introduction, citing recent work on virome ecology showing the predominant effect of host taxonomy on virus assemblages and transmission, and the contribution that ecological factors such as habitat overlap or inter-species interactions can have. We added the following sentences (lines 46-52):

“Studies primarily focusing on vertebrate hosts found a predominant role of host phylogeny on virome composition, and a limited, but sometimes significant effect of habitat overlap^{7,9-11} or inter-species interactions such as predator-prey relationships⁴ increasing cross-species virus transmission likelihood, particularly among phylogenetically related species^{12,13}. Here, we followed the temporal dynamics of insect pollinator viromes, exploring the relative roles of host phylogeny and plant-insect interactions on the dynamics of viruses.”

In the discussion, we now more strongly emphasize the role of host phylogeny in constraining virome composition in the light of our observation of limited cross-species transmission and replication from small RNA sequencing. We then discuss how increasing host diversity may limit cross-species transmission, and reads as follows (lines 277 to 298):

“Overall, these results illustrate how host phylogeny may deeply constrain the capacity of pathogens to infect a new host, and are consistent with previous work showing the effect of host taxonomy on virome composition in mammals^{9,13}, reptiles¹¹, birds¹⁰, fish⁶³, and insects^{5,64}, including bees^{24,44}. The high modularity of host-virus networks generated here illustrates further the specialization of viromes, and is consistent with previous studies⁴. The tight co-evolutionary relationships that enable cell entry and the use of the host cell replication machinery by viruses, before triggering an immune response, are likely to define the viral host spectrum, and may

have implications for viral ecology and wildlife conservation. For instance, there is a clear link between the current loss of biodiversity and the spread and emergence of infectious diseases. Lower host diversity has been shown to increase viral host jumps in various clades, such as coronaviruses in cave bat populations⁶⁵, and has been shown to be the global change driver most strongly associated with an increase in infectious diseases⁶⁶. This is also particularly relevant in the context of pathogen spill-over, which has been well documented from honeybees to wild pollinators^{27,37}, but for which the consequences in other insects remain poorly understood. The case of DWV, primarily infecting honeybees, but with potentially limited replication potential in bumblebees, is consistent with the dilution effect observed in areas with high pollinator diversity³³. Increasing bee diversity may thus provide a fragmented host landscape to those viruses that do not replicate equally well in all host species, potentially limiting their spread through dead-end spill over⁶⁷. It remains important to note, however, that viruses may also replicate in phylogenetically distant species like honeybees and bumblebees, as for example seen here for ABPV, a particularly virulent viral species. Indeed, at the individual level, we previously found evidence in this population for a dilution effect for DWV, but not for ABPV, where prevalence was driven by the abundance of its key host species instead³³.”

- Page 3 line 74 – 75. You need to include Latin names even if these are species groups. E.g. hoverflies are presumably Syrphidae spp.?

Response: *We agree and have added non-ambiguous taxonomic names in the revised manuscript.*

- In total RNA sequencing data there are often hits to divergent viruses e.g. ‘Riboviria sp’. What did you do with these viruses? Did you exclude them or classify them as insect or plant virus?

Response: *Indeed, we had 11 putative viral genomes mapping to unclassified viruses (e.g. Riboviria sp.) out of 143. For those, we could not resolve if they were viruses infecting the insect, their host plant or their microbes, and were categorized as ‘unknown host’ and discarded from the analysis. We have added this to our method description (line 403 in revised version).*

- Page 3 line 90. So were all vsiRNA signals that were 21nt fragments assumed to be plant viruses?

Response: *No, we did not rely on vsiRNA profiles to determine the original host. Thank you for identifying these confusing sentences and we apologize for the misunderstanding. We clarified our method in the revised manuscript, and modified the sentence to mention that we identified plant viruses based solely on sequence homology. Indeed, while a 21nt peak in vsiRNA profile from a bee sample is most likely the signature of a non-bee RNAi response (i.e. bee RNAi produces a majority of 22nt-long fragments as described by Remnant et al. 2017, doi:10.1128/JVI.00158-17), the reverse is not necessarily true: the plant RNAi response is diverse, with the production of profiles with peaks either at 21, 22 or 21/22nt (see Supplementary Figure S5 that shows vsiRNA profiles). Our correction can be found in the results section (line 97 to 99) that now reads as follows:*

“Transcriptome analysis resulted in the assembly and identification of 143 viral Operational Taxonomic Units (OTUs), including 39 plant viruses, identified via sequence homology.”

We also corrected our method section, that now reads as follows (line 411 to 419):

“Insects typically produce sense and antisense vsiRNAs of 21-23 nt long fragments⁸³. Bees produce vsiRNA profiles with a peak of 22 nt-long fragments⁴⁸, while plants generate peaks at different sizes (21 and 22nt)^{49,50}. [...] A positive siRNA response against viruses was determined with a minimum of 10 reads and a minimum coverage threshold of 5% of the target viral sequence, and was used to identify replicating viruses.”

- Page 3 line 100 – 101. Often you state a result and link to a figure, but there doesn't appear to be any statistics to back this up. For this sentence 'viromes restricted to insect viruses clustered mainly by host taxa' you could confirm this using a PERMANOVA analysis to determine if the viromes of different species were significantly different from one another. If you have actually done statistics on this, it needs to be made clearer. There are many other statements like this in the results where they don't appear to be backed up by statistics.

Response: *We agree that this particular claim was merely descriptive. To probe these findings with statistical tests, we performed PERMANOVAs on insect and plant virus distribution separately. For insect viruses, we found a strong effect of host taxonomic groups, and a small but significant effect of collection time point. Conversely, we found a stronger effect of collection time point on the virome composition in plant viruses, and host taxonomic groups, although to a lesser extent. We further supported this result by performing Non-metric Multidimensional Scaling (NMDS) projections of Bray–Curtis distance matrices. These plots, now in Figure 1B and 1C, illustrate our main results: the effect of taxonomic group on insect virus distribution and the effect of collection time point on the distribution of plant viruses in our libraries. These additional statistical results and figures are presented in the results and methods sections of the revised manuscript. We thank the reviewer for this suggestion.*

As for the other statements previously not fully backed up with statistics, we have added:

1) Two network analyses of modularity to demonstrate how only a limited number of viruses are exchanged across social species when considering the whole virome, and illustrate virome specialization using small RNA sequencing (see full answer to another comment below).

2) We clarified statistical outputs from our mixed-models as pointed out by reviewer #1 (see response to comment above), and

3) removed one statement about the observed increased dissimilarity in virome composition in plant viruses between insects with more distant ecological niches in June, that was not statistically significant ($p = 0.07$, see response to comment above).

- Page 5 line 129 – 132. Not convinced about this. In reference 38 the viruses they found infecting the bees had a distribution of 21-22nt long, not 22 only. A significant chunk of their vsiRNA for the bee viruses were 21nt long. In ref 39 I can't find anything saying the plant RNAis are 21 nt long, but in another paper (<https://www.mdpi.com/2073-4425/8/6/163>) you

can see there also some plant vsiRNA that is 21nt long. It isn't clear how exactly you distinguished between the plant and insect viruses, but I think you would need to look at the distribution of the vsiRNA sequence lengths rather than a strict 21/22nt cut-off. Also, Would these vsiRNA signals get degraded quicker than the virus genome? Is it possible that you are missing signals from infecting viruses because of this?

Response: *We are sorry if this sentence was poorly written. As mentioned above, we did not use vsiRNA profiles to determine the host taxonomic groups, precisely because of the diversity of profiles generated from plant RNAi. We corrected our manuscript to clarify this (see above) and changed our references (Tenllado et al. 2004 was an error) for Blevins et al. 2006 (doi:10.1093/nar/gkl886, reference #49) that demonstrated the role of the four Dicer-like (DCL) proteins in generating 21 to 24 nt-long vsiRNAs fragments in plants. We also added the citation of Niu et al. 2017 (doi:10.3390/genes8060163, reference #50) suggested by the reviewer.*

Concerning the question about small RNA degradation, we would argue that vsiRNA signals, if anything, are actually more resistant to degradation than whole viral genomes. The vsiRNA approach is known to be highly sensitive (see Roossinck et al, 2015; 10.1094/PHYTO-12-14-0356-RVW). For example, Wu et al. 2010 (10.1073/pnas.0911353107) showed that, in well-studied Drosophila and mosquitoes, additional RNA viruses could be discovered by using visRNAs. Experiments also showed that vsiRNA signals are clearly different from degradation signals: Marques et al. 2013 (doi:10.1371/journal.ppat.1003579) used Dicer-2 knock-out fruit flies (which cannot produce vsiRNAs) in comparison to wild-types, and found a clear difference of profiles and abundance of virus-derived small RNAs. In Figures 2 and 3 of their paper, one can see very few small RNAs in Dicer-2 knock-out flies experimentally infected with a virus, suggesting that random degradation is far less stable than true vsiRNAs observed in wild-types.

In our results (see Supplementary Material S5), typical bee vsiRNA profiles (i.e. with peak at 21 nt) are found for multiple infecting viruses, suggesting that our samples were not degraded. Therefore, we do not expect to be missing viruses from the vsiRNA because of degradation; rather, this might explain why we find some viruses in the vsiRNA (see comment on figure 2A below) that did not pass the threshold for detection in the meta-transcriptomes. To clarify this further (also in regards to the comment below), we have added the following in the text (lines 167-171):

“vsiRNA provides a highly sensitive detection method for viruses⁵¹, accordingly we detected seven viruses in the bumblebee vsiRNA data that were not found in the meta-transcriptomes (HLPV34 and New OLV, plus New Bee Iflaviridae 1; New Castleton Burn-like virus; New Jingchuvirales, New Rhabdoviridae 4; New Totiviridae 2) despite our conservative detection threshold.”

• Page 5 line 136. In the venn diagram I count 6 viruses that were found in vsiRNA signals, not 24. I think you've included ones found using vsiRNA in one species and the transcriptome in another which is not valid if you're assuming viruses are only multi-host if they are infecting both host species.

Response: *There are 24 viruses replicating in more than one host. The confusion comes from the Venn diagram discussed in the previous sentence, that represents the data at host genus level (Apis vs. Bombus). To clarify this, we added the mention “at the host species level” at the beginning of the sentence (line 157) and refer to the new supplementary*

material (Supplementary Material S4 & S5) that shows the list of replicating viruses in each sample (including coverage maps of small RNAs).

- Figure 2A. It looks like in the *Bombus* spp. you have 7 viruses that were only detected via vsRNA. How do you explain this?

Response: *Indeed, from 37 viruses detected in small RNA libraries from bumblebees, 7 were not detected in the meta-transcriptomes. These viruses are: HLPV34 and New OLV, plus New Bee I flaviridae 1; New Castleton Burn-like virus; New Jingchuvirales, New Rhabdoviridae 4; New Totiviridae 2. The reason for this lies with the detection threshold. We can see in Supplementary Material S4 (i.e. coverage maps from vsRNAs) that these viruses are present at low levels in the small RNA sequences. The same is true for meta-transcriptomes, and these viruses were simply not crossing the applied threshold. This illustrates nicely how these two methods may be complementary for virus discovery. See comment above for how we have changed the text to clarify this.*

- Figure 2B. This would be strengthened by network analysis (rather than just creating the network) e.g. community detection algorithms.

Response: *We agree that a formal network analysis would strengthen the support for our claim that these host-virus networks are compartmentalized, illustrating the host taxonomy effect on virome composition. To test this, we calculated the modularity Q of both meta-transcriptome and small RNA networks, and compared them to the respective computation of 100 randomized networks. In both cases, observed networks displayed significantly higher modularity than random networks. These results were added to the method (line 4331) and result (line 176) sections. We thank the reviewer for this suggestion.*

- Page 8. Discussion needs broadening - wider implications of our understanding of virus evolution, cross-species transmission, and comparison to other studies beyond studies of bees/plant-pollinator systems.

Response: *We thank the reviewer for this comment. We have broadened the discussion, particularly regarding the effect of host phylogeny being the most common factor influencing cross-species transmission of viruses, and its consequence for wildlife conservation in the light of biodiversity loss. This is now lines 277 to 298:*

“Overall, these results illustrate how host phylogeny may deeply constrain the capacity of pathogens to infect a new host, and are consistent with previous work showing the effect of host taxonomy on virome composition in mammals^{9,13}, reptiles¹¹, birds¹⁰, fish⁶³, and insects^{5,64}, including bees^{24,44}. The high modularity of host-virus networks generated here illustrates further the specialization of viromes, and is consistent with previous studies⁴. The tight co-evolutionary relationships that enable cell entry and the use of the host cell replication machinery by viruses, before triggering an immune response, are likely to define the viral host spectrum, and may have implications for viral ecology and wildlife conservation. For instance, there is a clear link between the current loss of biodiversity and the spread and emergence of infectious diseases. Lower host diversity has been shown to increase viral host jumps in various clades, such as coronaviruses in cave bat populations⁶⁵, and has been shown to be the global change driver most strongly associated with an increase in infectious diseases⁶⁶. This is also particularly relevant in the context of pathogen spill-over, which has been well documented from honeybees to wild pollinators^{27,37},

but for which the consequences in other insects remain poorly understood. The case of DWV, primarily infecting honeybees, but with potentially limited replication potential in bumblebees, is consistent with the dilution effect observed in areas with high pollinator diversity³³. Increasing bee diversity may thus provide a fragmented host landscape to those viruses that do not replicate equally well in all host species, potentially limiting their spread through dead-end spill over⁶⁷. It remains important to note, however, that viruses may also replicate in phylogenetically distant species like honeybees and bumblebees, as for example seen here for ABPV, a particularly virulent viral species. Indeed, at the individual level, we previously found evidence in this population for a dilution effect for DWV, but not for ABPV, where prevalence was driven by the abundance of its key host species instead³³.”

- Page 12 line 328. When was the genbank protein database downloaded?

Response: *The database was downloaded in February 22, 2021. This is now mentioned in the manuscript (line 394).*

- Page 12 line 333. 50 reads seem quite high. Maybe this is why you detected some vsRNA signals that you didn't detect in your transcriptome data? Have you tried looking for these missing viruses to see if they are present at a lower abundance? If so, assuming the vsRNA signal proves infection, it would suggest the 50 read limit is too high.

Response: *We agree that a threshold set to a minimum of 50 reads is rather high. We set this conservative threshold to reduce the number of false positives. We took this decision carefully and found that viruses with fewer mapping reads often also had a coverage rate below our threshold of 5%, increasing the risk of calling false positives with reads mapping to a tiny portion of the target genome sequence.*

Following the reviewer's concern, we looked specifically at the 7 viruses present in the bumblebee small RNAs and not present (or not passing the threshold) in bumblebee transcriptomes. For all of them, a rather low number of reads are found (<25 reads, see details below). In conclusion, the combination of transcriptome and small RNA sequencing allows the identification of more viruses, including viruses potentially present at low levels but efficiently recognized by the immune RNAi pathway of the host. To clarify in the manuscript that this represents a conservative threshold, we state it the sentence (lines 396-397):

“Each virus was considered to be present in a library if the number of reads was above a conservative threshold of 50, with a minimum coverage threshold of 5% and 250 nt of the target sequence.

To complete this response, we provide here the details of read numbers mapping to viral genomes in transcriptomes from samples showing positive detection small RNA sequencing:

Hubei partiti-like virus 34: 0 and 2 reads found in B. hortorum June and August transcriptomes, respectively.

New Bee Iflaviridae 1: 23 reads found in B. pascuorum (August) transcriptome.

New Castleton Burn virus: 6 and 6 reads found in B. pascuorum June and August transcriptomes, respectively.

New Jingchuvirales: 18 reads found in B. pascuorum (August) transcriptome.

New Osugoroshi-like virus: 8 reads found in B. hortorum (August) transcriptome.

New Rhabdoviridae 4: 8 reads found in B. pascuorum (August) transcriptome.

New Totiviridae 2: 8 reads found in B. pascuorum (August) transcriptome.

In addition, one virus, Castleton Burn virus, was found in the June honeybee small RNA sequences, but not considered positive in the transcriptome, as only 2 reads mapped to its genome.

Reviewer #3 (Remarks to the Author):

The manuscript entitled, “Host ecology and phylogeny shape the spatio-temporal dynamics of social bee viromes” by Doublet et. al Wilfert describes viral transcripts/RNA genomes and vsRNAs in RNA sequencing libraries obtained from social bees honey bees, and four bumble bee species, one solitary bee species (*Andrena* spp) and two flies (i.e., Hover flies and forgotten flies). The manuscript highlights some interesting and important findings including that of the ~24 insect viruses detected in multiple bee hosts in this study, only a five actively replicating viruses are likely shared between honey bees and bumble bees (i.e. ABPV, Castleton Burn virus, Phle1, Hubei partiti-like virus 34, and anew Osugoroshi-like virus) (i.e., based on vsRNA sequences).

It seems there is much more information that could be revealed by this dataset including, the full genome sequences and coverage maps of the main viruses detected and describe in this study, as well as the potential site and/or site/time specific differences in the viromes present in different bee species at the different sites, or at different sites and times.

Response: *We fully agree that there are many ‘hidden gems’ in this data set. Due to space limitations, we had to focus on ecological interactions and virome dynamics in the main manuscript. We have extended the supplementary material to provide additional data such as viral assemblies, coverage maps and siRNA profiles (see detailed responses to specific comments below). Unfortunately, we were not able to use barcodes for sites for our pooled RNA samples when generating libraries, therefore it is not possible to analyse viromes by sites.*

Points to clarify or address before publication include:

1. Abstract Line 25 – the word “sequences” should be added to the phrase “insect and plant virus sequences”.

Response: *We agree, we have changed the text to ‘viral sequences’ in the abstract (line 25).*

2. Abstract Line 25 – The text states that 6,000 insect samples were obtained, but the description of these samples is incomplete – and it seems the RNA from only a subset of these samples was pooled into 16 sequencing libraries (Supplemental Table S1). The sample list should include more details (e.g. geographic location/site, exact sample date, etc.). The authors should include more details about the number of each species at each site (i.e., 10 sites, x8 species, x 3 time points) that make up the pooled samples sequenced for this study. Another supplemental table should be included.

Response: We apologize for the lack of clarity. We corrected the number to 4,500 insects. We did collect ~6,000 insects for this project, but not all of the samples were used for RNA sequencing. Samples collected in spring 2017 were used for qPCR only and published in another paper on individual prevalence (Manley et al. 2023: doi:10.1098/rstb.2022.0004). In addition to Supplemental Table S1 that already provides the number of samples in each library, and referring to our previous publication that provides more information about our sampling regime, we now provide additional information in the current manuscript, with the dates of insect collection and general locations of sites in the method section. Unfortunately, for data protection of farmers that kindly accepted our presence on their land, we cannot provide more precise location data.

3. Abstract Line 28 – the word “diseases” should be replaced with “viruses”, or perhaps both words can be used.

Response: Agreed. Changed for the word ‘virus’.

4. Lines 208, 210 and throughout the manuscript “virus titer” should be changed to “virus RNA copy number”, as “titer” is typically used to indicate virion number as assessed by plaque assays and the data presented in this paper is based on RNA sequence abundance (i.e., virus RNA copy number per xxxx ng RNA), which includes both genomic RNA and transcripts. Relative virus abundance may be ok too.

Response: We agree that our measure of RNA viral copies is not a precise measure of viral titer. Thus, we changed for ‘loads’ in now lines 241 & 245. For line 243, the mention of titer referred to a general comment from the literature, and we therefore decided to keep this term in this instance.

5. Lines 196-197 – Consider using “transmission” or “spread” rather than spillover
The authors should use “transmission” rather than “spillover/spillback” since many studies have shown that BQCV (and DWV) are prevalent in both honey bees and bumble bees, so these viruses are likely just shared between bee species within a community. Viruses are transmitted between different genera / species of bees – so that use of the word “spillover” is not always appropriate. Although, there may be several definitions of the word “spillover” (e.g., Wikipedia “Spillover infection, also known as pathogen spillover and spillover event, occurs when a reservoir population with a high pathogen prevalence comes into contact with a novel host population. The pathogen is transmitted from the reservoir population and may or may not be transmitted within the host population.” doesn’t seem accurate for bee viruses. Additional temporal studies are required to determine the ecology of bee viruses – as well as to describe these events as either potential spillover or spill back.

Response: Here, we would like to disagree with the reviewer’s point that the terms spillover and spillback do not apply to bees. On the contrary, this concept is largely accepted by the community as exemplified by the growing number of papers referring to the spillover of viruses in their title: e.g. Manley et al. 2019 [10.1111/ele.13323], Dalmon et al. 2021 [10.3390/insects12020122]; Fleites-Ayil et al. 2023 [10.1016/j.biocon.2023.110150], Santamaria et al. 2018 [10.1016/j.jip.2017.11.008], Pislak Ocepek et al. 2021 [10.3390/pathogens10070884]. As defined by reviewer #3 (above, citing Wikipedia), spillover happens from reservoir species (i.e. primary species) with high disease prevalence to secondary host species that then displays lower prevalence. This is precisely what is observed between sympatric honeybees and bumblebees in several recent publications with

these two viruses (for BQCV: McMahon et al. 2015 [10.1111/1365-2656.12345], Manley et al. 2020 [10.1111/mec.15333], Dalmon et al. 2021 [10.3390/insects12020122], Pislak Ocepek et al. 2021 [10.3390/pathogens10070884]; and for DWV: Fürst et al. 2014 [10.1038/nature12977], McMahon et al. 2015 [10.1111/1365-2656.12345], Manley et al. 2023 [10.1098/rstb.2022.0004], Manley et al. 2019 [10.1111/ele.13323], Dalmon et al. 2021 [10.3390/insects12020122], Pislak Ocepek et al. 2021 [10.3390/pathogens10070884], Döbelmann et al. 2024 [10.1098/rsbl.2023.0600]).

6. Line 224 – “disease” should be replaced with the word “virus” – since DWV is a virus, not a disease.

Response: *Agreed. We changed for the word ‘virus’ (now line 261).*

7. For the main viruses discussed (DWV-A, DWV-B, ABPV, SBPV, Castleburn virus, and others), the authors should map their sequencing date back to the viral genomes and describe the depth of sequence coverage, overall genome coverage, percent identity, etc. This data should be included as figures or tables (maybe one main figure, and supplemental info). The virus genome sequencing data for all of the described viruses should be deposited on NCBI and accession numbers added to the text. The percentage identity shared with the most similar genome on NCBI should also be listed.

Response: *We would like to thank the reviewer for these suggestions. We now provide the sequence assemblies for the full and partial genomes we reconstructed, available in a repository (link at the end of the manuscript). However, we do not consider these suitable for publication in NCBI GenBank (indeed, they would probably not be accepted), as these genomes were generated from a pool of samples, very likely leading to chimeras of strains present in our sampled host populations. Because of this, we also do not consider further descriptive data, particularly the percentage identity to currently published viral strains, to be informative (although if the editor disagrees, we are happy to provide this as additional supplementary material). Instead, we focused on the coverage analyses from small RNA sequencing (see below). We have now generated coverage maps of vsiRNA libraries per species and collection time points (Supplementary Material S4). This representation of data provides valuable and insightful information about both viral replication and the temporal dynamics of single viruses in our samples. We have added mentioned of these new representations in the result section, e.g. line 159.*

8. The authors could include more detail and description about the sequences in each library (e.g., were the five many viruses detected in all libraries generated from all sample dates, were all of those viruses detected at all 10 sites (if samples were tagged by site) The methods section indicates 17 libraries were sequenced by the text states 16. In addition, to the number of total reads obtained for each sequencing library, the authors should include more analyses information in the Supplemental Table S1 (e.g., the number/percentage of reads that aligned to viruses in each library, the reads assigned to specific key viruses in each library, etc.). The numbers should have “,” rather than decimal places. It would be great if the authors could share more of their sequence data in an analyzed format (e.g., virus sequences binned by virus, etc.).

Response: *Unfortunately, we could not afford tags per sites, it is therefore impossible to comment about the presence or abundance of viruses across sites and single samples from this data. In our previous publication (Manley et al. 2023: doi:10.1098/rstb.2022.0004), we*

however analysed prevalence and abundance of three key viruses (deformed wing viruses A and B, and acute bee paralysis virus) at the individual level, including site and time point. To highlight this, we modified the first sentence of our discussion (lines 217-220), that now reads as follows:

“In this study, we combined comparative meta-transcriptomics, small RNA sequencing and ecological network analysis to identify drivers of virome composition in insect pollinators, following on from our study of the transmission dynamics of three key bee viruses (DWV-A, DWV-B and ABPV) at the individual level in this population [Reference #33, Manley et al. 2023].”

Regarding the number of libraries, we initially indeed generated 17 libraries, both for the meta-transcriptomes and small RNA sequencing. All of these were deposited in GenBank. However, we reduced our analysis to 16 and removed one bee species pool composed of a mix of phylogenetically distant species and including less abundant bumblebees together with halictid bees and nomad bees. We included this pool for experimental reasons to explore the viral diversity in the community of insect pollinators of our study sites. However, it is not relevant to the analysis in this study as this pool would have had similarities with other bumblebee single-species meta-transcriptomes due to the presence of bumblebees in the mix, but also similarities to mining bees, due to the presence of non-apidae samples. To avoid confusion and inconclusive associations in our analysis, we decided to not analyse this sample here. We clarified this point by correcting the number of analysed transcriptomes to 16 in the manuscript.

We modified our table in Supplementary material S1 and used commas to make large numbers more readable, and added the percentages of reads mapping to viral targets for both meta-transcriptomes and small RNA sequences. We also made available the sequence of viral assemblies in a repository, while the full meta-transcriptomes are available on Genbank.

Finally, as requested here, we have made available all our screening results from meta-transcriptomes and small RNA sequencing, including the number of viral reads binned per viral assemblies, in a repository (link available at the end of the manuscript).

9. Depending on how much of the Castleburn virus the authors assembled (e.g., only RdRp sequence vs. entire genome), it may be that they are detecting a recently described virus (i.e., *Andrena* associated bee-virus 1 (AnBV1). AnBV-1 has a bipartite RNA genome (RNA (MW397641.1 – 2,721 bp with RdRp and MW397640.1 RNA, 2005 bp) with the RdRp containing RNA sharing regions of high identity to Castleburn virus (GenBank: MH614293.1 2,714 bp)(BLAST nucleotide alignment 4e-30 246/363 - 68% identity). see Daughenbaugh et al, *Viruses* 2021, 13(2), 291; <https://doi.org/10.3390/v13020291> “The putative 499 aa AnBV-1 RdRp sequence produces the strongest alignment with a putative Castleton Burn virus RNA-dependent RNA polymerase, (BLASTp e-value = 0), which was identified by sequencing bumble bee samples (Supplementary Figure S11) [156]. The RNA-dependent RNA polymerase proteins of AnBV-1 and Castleton Burn virus are similar. However, they share only 53.3% amino acid identity (Supplementary Figure S11).”

Response: *Both viruses are indeed close, but clearly identifiable: their conserved RdRp amino acid sequences are similar at 58%, as shown in Supp Mat S11 of Daughenbaugh et al. 2021. Using our Castleton Burn contig sequence, we find the same result: 95% nucleotide identity with the sequence MH614293.1 from Pascall et al. (2019,*

10.1101/498717), and 97% identity with its translated RdRp protein. In contrast, there is only 58% identity to the translated RdRp from AnBV1 over the same ~1500 nt long sequence. This confirms that our contig belongs to Castleton Burn virus and not AnBV1.

10. Figure 2A – should include a table with the shared virus names (~ 36 viruses), the 2, 2, and 6 shared viruses could be listed on the figure, and the others should be included in a supplemental table.

Response: Yes, we agree that this figure could provide more information on the shared viruses. As suggested, we added the abbreviations (explained in the figure caption) of the 5 viral species found replicating both in honeybees and bumblebees: ABPV, *Bombus* associated virus Phle1, Castleton Burn Virus, Hubei partiti-like virus 34 and New Osugoroshi-like virus. The names of the other viruses, shared between honeybee and bumblebee transcriptomes, but not found in both group's siRNA libraries, are now available as a table in Supplementary Material S3.

11. Line 163 “pollinator-associated or bee-associated” should be added before the words “plant viruses” to clarify the meaning of this sentence.

Response: Agreed. We added 'bee-associated' to clarify the meaning (now line 197).

12. Line 191, should be rephrased. This study involved sequencing of insect associated viruses at three distinct times in Scotland, and the data included are insufficient to state that “many insect viruses not able to jump from one host to another”. It would be better to state that “not many insect viruses (or virus sequences) were shared between species in this study, which may indicate host specificity”.

Response: Yes, we agree, it is not possible to be that conclusive on the capacity of viruses to jump hosts with our dataset alone. We rephrased it as suggested. It now reads (now lines 223-226):

“If we consider plant-pollinator networks as a proxy of virus transmission potential, meta-transcriptomes suggest that not all of this potential is realized as only half of the identified insect viruses were shared between species in this study, indicating some degree of host specificity”.

13. The vsiRNAs should be mapped back to select virus genomes to illustrate virus genome coverage, and provide additional support for replicating viruses. It is great that the authors describe this data, but the figures do not well-represent these findings.

Response: We agree that additional plots will improve the comprehension and visualization of this complex dataset. We followed the reviewer's advice and now provide the coverage maps of vsiRNAs of viruses (passing a threshold of 5% coverage overall) for each taxonomic group, and colored by collection time points. This is now available as Supplementary Material S4, with the method described in the main text (lines 420-422). We can clearly see the highest number of viruses replicating in honeybees, and the temporal dynamics of infection with coverage depth. This can also be read as a list of replicating viruses in our samples. We would like to thank the reviewer for this suggestion.

14. Line 125 – change the word “pathogens” of “multi-host pathogens” since this study only

demonstrates that these are bee-associated virus sequences, it would be better to just call them “viruses” – since the authors did not demonstrate pathogenicity.

Response: *Agreed. The mention of ‘multi-host pathogens’ here might have been misleading. We removed the word pathogen and it now reads (now lines 157-159):*

“... we found 24 insect viruses (25%) to be multi-host using this method, i.e., with vsiRNA evidence of replication in more than one host species.”

Minor points to clarify or address before publication include:

1. Key words “honey bee” and “bumble bee” are two words, whereas they are written as compound words throughout the manuscript.

Response: *Yes, we used the terms “honey bee” and “bumble bee” for referencing purpose, as these words are also widely used in this form in the scientific literature, but decided to use compound words, which is standard practice in British English, for the article.*